# Skeletal muscle transcriptional dysregulation of genes involved in senescence is associated with prognosis in severe heart failure

Eric Rullman ✉, Alen Lovric , Michael Melin , Rodrigo Fernandez-Gonzalo & Thomas Gustafsson

## Abstract

**Background** The skeletal muscle hypothesis refers to a vicious cycle of successive deterioration of left ventricular function, skeletal muscle remodeling, and functional capacity in patients with heart failure. Despite extensive research, the regulatory mechanisms and their associations with clinical status and prognosis are still largely unclear.
**Methods** To identify mechanisms and characterize underlying processes involved in the disease pathophysiology, we performed RNA sequencing and network analysis using human skeletal muscle samples from 58 patients with severe symptomatic heart failure. A co-expression network with communities involved in established biological processes within human skeletal muscle was identified and validated in two independent cohorts.
**Results** Here, we show network communities associated with mitochondrial beta-oxidation, extracellular matrix remodeling, oxidative phosphorylation, and contractile elements with lower expression in heart failure patients than in age-matched controls. Based on the strong correlation with clinical features and prognosis, extracellular matrix remodeling, mitochondrial beta-oxidation, and p53 signalling communities are identified as key underlying processes. The former two communities are highly enriched with genes regulated by physical (in)activity, i.e., bed rest and exercise, and associated weakly with prognosis. Community related to p53 signalling, with *CDKN1A* as a key regulator, is increased in heart failure patients relative to age-matched controls and associated with worse prognosis.
**Conclusion** The current work differentiates previously proposed factors underlying heart failure-induced skeletal muscle dysfunction, emphasizing the p53 signalling community and importance of biological age in this process. The distinct association with clinical status and prognosis furthermore supports pathophysiological significance and clinical potential of this community.

## Plain language summary

Heart failure patients suffer from muscle weakness and reduced physical activity, symptoms that worsen with the disease progression. Although extensively studied, biological mechanisms behind it remain largely unclear. Here, we analysed skeletal muscle samples from patients with severe heart failure using gene-expression-based approach where transcripts with similar expression are grouped together into networks. We identified muscle processes related to energy metabolism and tissue repair to be altered in these patients, likely in part due to their physical inactivity. Importantly, we identified increased expression of genes linked to p53 signalling, process associated with aging and cell damage, which was also strongly related to patient outcomes. Our results distinguish and highlight the need to separate the negative effects of physical inactivity from disease-specific changes, which is essential for discovering new treatment targets.

Heart failure with reduced ejection fraction (HFrEF) is a common condition with an estimated prevalence of up to 3% of the adult population and a remarkably high mortality rate (35–50% in symptomatic patients)[1,2]. The main manifestation of heart failure is limitation of physical performance due to dyspnea and fatigue[3–5]. These symptoms worsen in parallel with the severity of the disease and are among the most crucial factors in the deterioration of the quality of life of heart failure patients[6]. Interestingly, skeletal muscle structure and function become increasingly important for physical performance as the disease progresses[4,7,8]. Physical performance, including peak oxygen uptake ($VO_{2peak}$), has repeatedly been shown to be a prognostic marker in heart failure[5,9]. In addition, handgrip strength and muscle mass, for example, have been shown to predict disease progression and prognosis[10].

Considering the apparent interaction between skeletal muscle and heart failure, researchers and clinicians have coined the term "skeletal

Division of Clinical Physiology, Department of Laboratory Medicine, Karolinska, Institutet, and Unit of Clinical Physiology, Karolinska University Hospital Huddinge, Stockholm, Sweden. ✉e-mail: eric.rullman@ki.se

muscle hypothesis" to describe the vicious cycle between myocardial function and skeletal muscle remodeling that leads to successive deterioration of left ventricular function and functional capacity[4,11]. So far, a number of case-control studies have established heart-failure related changes in fiber-distribution, capillary density and mitochondrial content. However, it is still unknown to what extent the changes associated with heart failure are disease-specific[12–14], cachexic processes observed also in other diseases, the effects of physical inactivity or simply the effects of general aging[8,15,16]. Indeed, a deeper understanding of the mechanisms underlying the skeletal muscle hypothesis in heart failure could lead to better strategies to counteract the adverse effects of the disease.

An alternative approach to gain insights into the underlying pathophysiology of HFrEF is to examine the complex relationships between the molecular processes associated with skeletal muscle dysfunction and their impact on prognosis. For example, identifying an enrichment of genes disrupted in heart failure and affected by variations in physical activity would underscore the vital role of muscle dynamics in disease progression. In addition, information on how different skeletal muscle features in heart failure relate to other key aspects of the disease would help identify underlying regulatory stimuli and could also be used to search for therapeutic targets. In this context, high-throughput transcriptomic profiling enables data-driven exploration of the patient's condition and, as the number of observations increases, analysis of features and correlations that go beyond the simple case-versus-control design. One such technique, co-expression analysis, has proven to be a powerful tool for systematically deciphering molecular responses and/or identifying critical signalling pathways relevant to phenotypic traits or disease prognosis[17]. Network-based analysis strategies often prove superior to traditional case-control designs because of their ability to capture complex relationships and interactions within a system.

Here, we present a co-expression network constructed based on RNA sequencing of muscle samples from 58 patients with severe symptomatic HFrEF. The co-expression network and underlying communities are validated in multiple population-based datasets with >600 skeletal muscle transcriptional profiles. The co-expression network is (i) compared with age- and comorbidity-matched individuals without heart failure and (ii) analyzed with respect to a variety of clinical characteristics and end points, as well as prognosis in terms of all-cause mortality, both in the 58 patients included in the study and in independent datasets.

The current work distinguishes previously proposed factors underlying heart failure-induced skeletal muscle dysfunction, emphasizing the p53 signalling community and the importance of biological age in this process. The distinct association with clinical status and prognosis further supports the pathophysiological significance and clinical potential of this transcriptional dysregulation.

## Methods
### Study participants
Heart failure patients who met inclusion criteria (NYHA III-IV and left ventricular ejection fraction ≤30%, age 60-75 years) and were willing to provide a skeletal muscle biopsy and participate in clinical characterization, including VO$_{2peak}$ and daily activity measurements, were enrolled in the study over a 3-year period (Fig. 1). Patients were followed prospectively for an average of 3.5 years (range 1–5). In parallel with patient recruitment, a control group of 20 participants was recruited. Control participants were included in the study after being referred to the cardiology department by general practitioners with signs and symptoms of heart failure, but echocardiography revealed normal systolic function, normal natriuretic peptides and without echocardiographic evidence of elevated filling pressure (E/e' <14, LA-area <20 cm$^2$ or LAVI < 34 ml/m$^2$ and TR-Vmax <2.8 m/s) i.e., they did not meet the criteria for heart failure (Fig. 1). Routine echocardiography, skeletal muscle and blood sampling, and a symptom-limited cardiopulmonary exercise test with measurement of VO$_{2peak}$ were performed in all participants. Continuous assessment of gas exchange data (Vmax, SensorMedics, Anaheim, CA, USA) was performed during the

exercise test. Echocardiographic measurements were performed according to clinical guidelines (Vivid 7, General Electric's Healthcare, Little Chalfont, United Kingdom) and evaluated by an echocardiologist blinded to the patient's specific clinical history. Left ventricular end-diastolic diameter (LVEDD) was measured and left ventricular ejection fraction (LVEF) was calculated using biplane Simpson. Left ventricular filling pressure was estimated from E/e'. Daily activity was determined by continuous accelerometry over 7 days. Patients were instructed to carry the accelerometer (GT3X; Actigraph, Pensacola, FL, USA) in their waist belt always except for showering, bathing, and sleeping. Baseline data on mortality and cause of death in the 12-56 months (about 4 and a half years) after testing were obtained from the Swedish National Registry of Causes of Death. To ensure external validity of the group of patients recruited baseline characteristics of all patients with HFrEF aged 60-75 years in NYHA III-IV registered at the outpatient clinic at the Karolinska University hospital ($n = 328$) during 2015 was obtained and compared with the studied cohort. To minimize the risk of gene-expression differences between patients and controls being driven by confounding factors, an additional control-group was generated by means of propensity-matching from the population-based Gene Tissue Expression Dataset (GTEx; https://gtexportal.org/home/). A balanced (i.e., $n = 58$) control group was pulled from the 515 individuals after propensity scoring based age, sex, presence of diabetes, and body mass index.

### Tissue sampling and RNA sequencing
Blood samples were collected in a fasted state in the morning using EDTA-coated tubes. The tubes were immediately centrifuged, and plasma was stored at −80 °C until analysis. Skeletal muscle biopsies were performed in m. vastus lateralis with a 5-mm Bergström needle with suction applied. RNA was isolated from muscle tissue using Trizol (Thermofisher) and subsequently purified using RNAEasy columns (Qiagen). RNA concentration and integrity were analyzed using Bioanalyzer (Agilent). Trueseq Poly-A enriched libraries (Illumina) were prepared from 200 ng of total RNA and libraries were pair-end sequenced using Illumina Nextseq 550. Raw sequencing reads were submitted to FASTQCR (v0.1.2) for quality control and aligned to human reference genome (GRCh38.92) using STAR aligner (v2.7.9a). Gene counts were summarized using featureCount (v1.5.1). After trimmed mean M normalization, expression matrices were converted to log$_2$CPM values using edgeR.

### Identification of a reproducible skeletal muscle transcriptome
Gene expression in the heart failure cohort was confirmed in muscle tissue from the GTEx and from the Studies Targeting Risk Reduction Interventions through Defined Exercise (STRRIDE)[18] (Fig. 2). The GTEx dataset is the largest human skeletal muscle expression dataset available to date and consists of 515 RNA sequencing libraries derived from different muscle groups. The age range is broad (20–70 years), gender is balanced, and various ethnic groups are represented. Genes were considered expressed in skeletal muscle in the GTEx dataset if they mapped >1 counts in at least 20% of libraries. The STRRIDE study (pre-intervention data) was selected as an additional reference dataset due to its size ($n = 125$ participants) and similarity to the present cohort in terms of age and comorbidities. The STRRIDE study was analyzed with Affymetrix U133 microarrays. Uninformative genes in STRRIDE dataset were identified using inter-quartile range (IQR) cutoff <2 and excluded from further analysis.

### Construction of a heart failure weighted gene co-expression network
Weighted gene co-expression network analysis (WGCNA) was used to construct a soft threshold co-expression network (Fig. 2). A threshold of 13 was used to obtain a scale-free topology and highly similar communities were merged based on a cut-height of 0.25 in the hierarchical clustering. To minimize the risk of co-expression being driven by spurious correlations, all edges were recalculated in the GTEx dataset and in the STRRIDE study. Communities of transcripts detected in the heart failure patients were validated using the module-preservation feature of the WGCNA

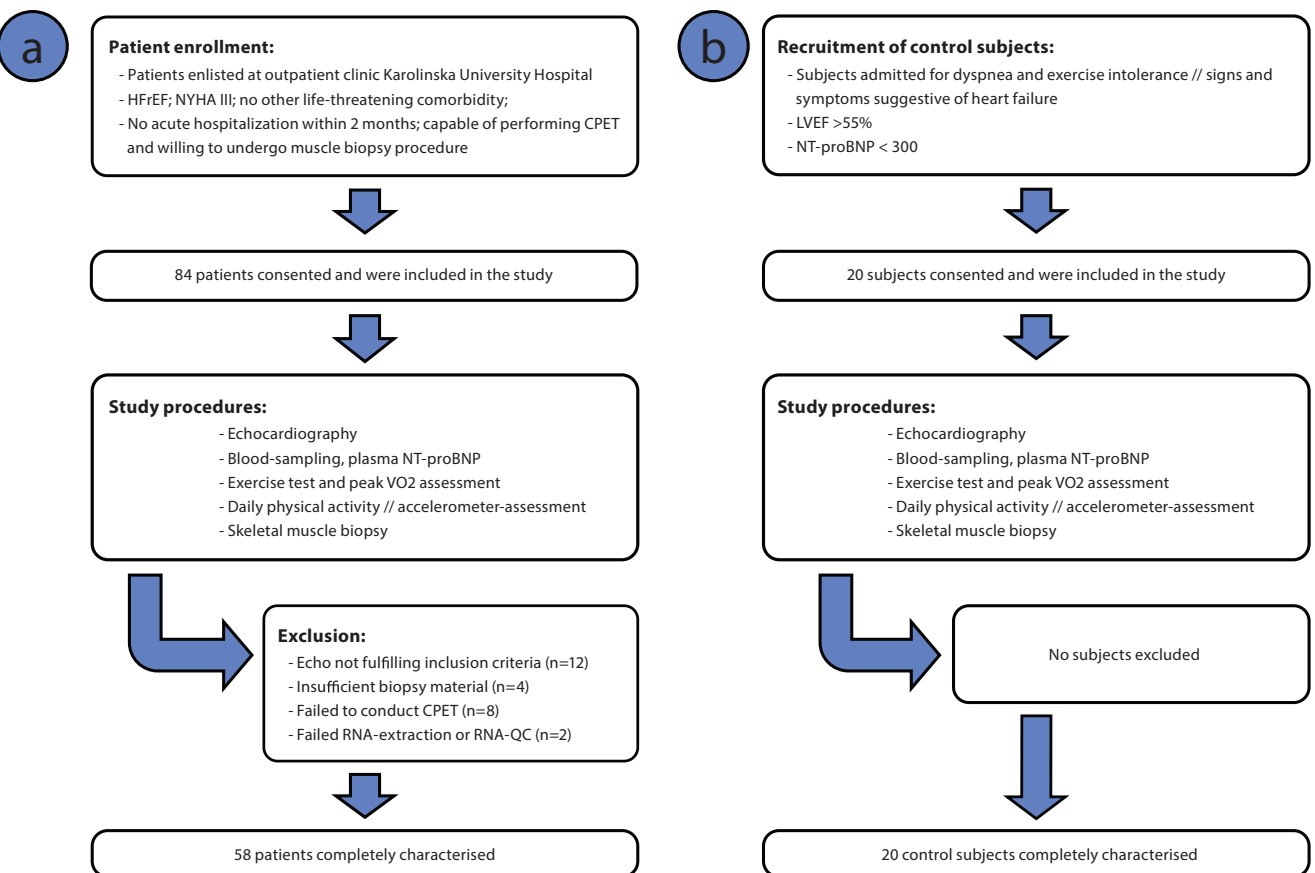

**Fig. 1 | Flowsheet of patient and control-subject recruitment, characterization, and follow-up. a** Heart failure patients with HFrEF, clinically stable with severe exercise intolerance (NYHA III) but with no life-threatening comorbidities were asked to participate in the study at the outpatient clinic of the Karolinska University Hospital. In total, 84 patients were willing to give up a skeletal muscle biopsy and undergo all diagnostic procedures associated with the study and were followed for 60 months (about 5 years) regarding all-cause mortality. Of those, 12 patients did not fulfill echocardiographic inclusion criteria, 14 patients dropped-out due to not being able to undergo all clinical characterization or issues with skeletal muscle sampling,

RNA-extraction, or quality. **b** A control group was recruited in parallel with heart failure patients from individuals admitted to the clinic with signs and symptoms indicative of heart failure but did not meet criteria of heart failure based on the echocardiography and normal NT-proBNP levels (LVEF > 55% and NT-proBNP <300 ng/ml). Control group underwent the same clinical characterization as the heart failure patients with HFrEF. The external validity of the final study group of patients was assessed by comparison of clinical baseline characteristics of the study group with records from all patients with HFrEF enlisted at the clinic throughout the study period (Supplementary Table 1).

methodology. The Z-score metric of all communities was calculated in STRRIDE and GTEx, and communities classified as at least moderately preserved (i.e., Z-score ≥3) in ≥1 of the datasets were retained in the network.

### Biological characterization of network communities

Biological characterization through gene ontology and pathway enrichment analysis was performed with clusterProfiler, testing each network community using the 8867 network transcripts as background, and an FDR of <0.05 was considered significant enrichment (Fig. 2). Annotation data for biological processes were obtained from Broad Institute c5.bp.v6. Pathway annotation data were obtained from Wikipathways v7.4, and metabolic pathway annotations were obtained from Human-GEM 1.10.0.

### Network community gene-expression compared to control participants, correlation with clinical traits, and prognosis

Gene expression profiles of network communities were summarized using principal component analysis (Fig. 2). The first principal component (eigengene) representing an aggregated expression profile, was used to associate community-level expression with clinical characteristics and to assess community expression differences between heart failure patients and controls. Pearson's correlation was used to assess associations with continuous clinical variables, and Welch's t-test was applied to evaluate group

differences. A nominal $P$-value of <0.05 was considered statistically significant. The network communities were also analyzed for over-representation of genes associated with several clinical conditions relevant to heart failure. Here, the gene-set collections were created using differentially expressed transcripts reported by the respective original publications: (1) Transcripts regulated by increased physical activity/exercise training was extracted from[19] where an FDR < 0.05 in pre- versus post-exercise training was considered an exercise regulated gene. (2) Long-term physical inactivity in otherwise healthy individuals in the form of 90 days bed rest[20] (a differential expression with FDR of <0.05 was considered a bed-rest regulated gene). Cancer cachexia in the form of a skeletal muscle transcriptional profiling of patients undergoing surgery for pancreatic cancer in contrast with healthy controls[21] (a nominal $P$-value of <0.05 was considered a cachexia-regulated gene). Finally a gene-set of transcripts involved in cellular senescence was obtained from the CellAge project[22]. Over-representation analysis was performed on individual communities in heart-failure network with Fisher exact test using transcripts from the rest of the network as background, and a nominal $P$-value of <0.05 was considered significant enrichment. A complete list of transcripts used to create four gene-sets are provided in Supplementary Data 1. Association with prognosis was tested using co-expression with death regardless of the cause as dependent variable, and with the aggregated gene-expression of each network community divided into terciles (i.e., low, intermediate, and high

## Analytical work-flow:

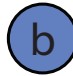

**Heart failure patients RNA-sequencing:**
In total 13 486 transcripts mapped

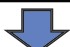

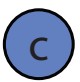

**Validation of skeletal muscle expression:**
- Transcript expressed in skeletal muscle of subjects in GTEx or STRRIDE?

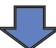

In total 8867 transcripts with validated skeletal muscle expression

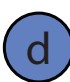

**Weighted gene co-expression network analysis:**
19 tentative communities of highly correlated transcripts

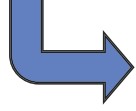

**Network validation and pruning:**
- Validation of co-expression in GTEx or STRRIDE
- Confirmation of network communities in GTEx or STRRIDE

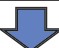

Final network consisting of 1481 transcripts with >1 million validated correlations, and across 14 distinct communities

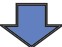

**Biological characterization of network communities:**
- Gene-ontology, pathway, and genome-scale metabolic model analysis
- Transcriptional regulation by exercise training or deconditioning though prolonged bed rest
- Transcriptional regulation in cancer cachexia
- Enrichment of genes associated with cellular senescence

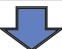

**Skeletal muscle expression of network communities in heart failure:**
- Differential regulation in heart failure patients compared with healthy controls (n=20), and propensity-matched GTEx individuals (n=58)
- Transcriptional regulation in relation to clinical traits and prognosis
- Validation of gene-expression in relation to prognosis and heart disease in GTEx

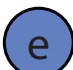
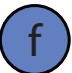

## Validation datasets:

**Studies Targeting Risk Reduction Interventions through Defined Exercise (STRRIDE) study:**
- Skeletal muscle microarray analysis
- n = 125
- Age 24-68 years

**Gene tissue expression dataset (GTEx):**
- Skeletal muscle RNA sequencing analysis
- n = 515
- Age 18-86 years
- 42% Females
- 12% died from heart disease

**Fig. 2 | Skeletal muscle transcriptional analysis workflow.** Transcripts detected in the muscle samples originating from the heart failure patients (**a**) and where skeletal muscle expression could be confirmed in either the STRRIDE or GTEx cohorts (**b**) constituted the starting point for the co-expression analysis. Initial network was thereafter constructed based on weighted co-expression of transcripts in the heart failure cohort and the network was organized into communities based on the connectivity (i.e., number of significant correlations) between transcripts across patients (**c**). To ensure external and biological validity, and to minimize the risk of false discoveries, the initial network was pruned from edges (i.e., gene-gene correlations) that could not be confirmed in the independent skeletal muscle datasets from STRRIDE and GTEx. The validity of the communities in the network was also confirmed by calculation of module-preservation statistics and communities of the network that was not preserved in the independent cohorts were discarded from further analysis (**d**). The communities of the final network were characterized through gene-ontology, pathway and genome-scale metabolic modelling and was named based on highly enriched GO-terms. Each community was also tested for regulation in published datasets on clinical conditions with potential relevance for heart failure; Exercise training, prolonged bed rest, cancer cachexia and cellular senescence (**e**). Finally, the network communities were analyzed for regulation in heart failure patients in relation to the control-subjects (*n* = 20) and propensity-matched (on age, BMI, and comorbidities) individuals from GTEx (*n* = 58), and in relation to clinical traits and prognosis (**f**).

**Table 1 | Detailed demographic and clinical characteristics of study participants**

| | Control subjects (N = 20) | Heart failure (N = 58) | P-value |
|---|---|---|---|
| **Baseline** | | | |
| Sex: Female | 12 (60.0%) | 11 (19.0%) | 0.001 |
| Male | 8 (40.0%) | 47 (81.0%) | |
| Age | 70.5 [65.8, 73.0] | 71.0 [63.3, 74.0] | 0.832 |
| BMI (kg/m$^2$) | 25.0 [23.0, 27.0] | 28.0 [25.0, 30.0] | 0.028 |
| Systolic blood pressure (mmHg) | 145 [135, 160] | 120 [110, 135] | $5.54 \times 10^{-05}$ |
| **Exercise** | | | |
| Time spent physically active (%) | 33.7 [31.2, 38.7] | 20.2 [16.8, 28.3] | $1.67 \times 10^{-04}$ |
| peakVO$_2$ (ml/kg/min) | 25.4 [21.9, 29.4] | 14.6 [12.0, 17.8] | $7.04 \times 10^{-07}$ |
| **Blood** | | | |
| NT-proBNP (ng/l) | 94.5 [51.3, 239] | 1990 [934, 4850] | $8.45 \times 10^{-09}$ |
| eGFR (ml) | 69.4 [58.7, 79.8] | 61.2 [45.6, 81.1] | 0.491 |
| Hemoglobin (g/L) | 139 [133, 144] | 141 [127, 153] | 0.595 |
| **Echocardiography** | | | |
| LVEF (%) | 59.0 [55.0, 60.0] | 25.0 [20.0, 31.0] | $3.03 \times 10^{-31}$ |
| PA-pressure (mmHg) | 25.0 [0, 30.0] | 41.0 [0, 52.0] | $3.42 \times 10^{-04}$ |
| LVEDD (mm) | 44.5 [42.0, 50.5] | 64.0 [59.0, 69.0] | $3.38 \times 10^{-12}$ |
| TAPSE (mm) | 22.0 [18.0, 25.0] | 14.0 [11.0, 18.0] | $3.91 \times 10^{-07}$ |
| E/e' | 8.88 [7.98, 10.9] | 18.3 [14.0, 23.7] | $1.39 \times 10^{-09}$ |
| Left atrial area (cm$^2$) | 17.5 [15.0, 20.0] | 30.0 [26.0, 36.0] | $5.69 \times 10^{-15}$ |
| **Comorbidities** | | | |
| Atrial fibrillation | 0 (0%) | 31 (53.4%) | $7.22 \times 10^{-06}$ |
| Diabetes | 1 (5.0%) | 25 (43.1%) | 0.002 |
| Hypertension | 12 (60.0%) | 34 (58.6%) | 1 |
| **Pharmacological treatment** | | | |
| RAAS blockade | 7 (35.0%) | 54 (93.1%) | $5.44 \times 10^{-07}$ |
| Betablockers | 6 (30.0%) | 55 (94.8%) | $1.95 \times 10^{-08}$ |
| Mineral receptor antagonist | 0 (0%) | 37 (63.8%) | $1.44 \times 10^{-07}$ |
| **Device therapy** | | | |
| CRT | 0 (0%) | 21 (36.2%) | $8.86 \times 10^{-04}$ |
| ICD | 0 (0%) | 36 (62.1%) | $4.46 \times 10^{-07}$ |
| **Heart failure etiology** | | | |
| Dilated cardiomyopathy | | 38 (65%) | |
| Ischemic heart disease | | 25 (43%) | |

A two-sided Welch's two-sample t-test was used for continuous variables and Fisher's exact test was used for categorical variables. Data are presented as median (IQR) or n (%). Exact nominal P-values are reported in the table.

Abbreviations: *BMI* body mass index, *NT-proBNP* N-terminal pro-B-type natriuretic peptide, *eGFR* estimated glomerular filtration rate, *LVEF* left ventricular ejection fraction, *PA* pulmonary artery, *LVEDD* left ventricular end-diastolic diameter, *TAPSE* tricuspid annular plane systolic excursion, *E/e'* ejection fraction (E)/early diastolic mitral annular velocity (e') ratio, *RAAS* renin-angiotensin-aldosterone system, *CRT* cardiac resynchronization therapy, *ICD* implantable cardioverter-defibrillator.

expression) as independent variable. A nominal Wald *P*-value of <0.05 was considered as a significant association with prognosis.

The GTEx dataset was used to validate the relationship between expression of transcriptional communities and prognosis. In this analysis, the transcriptional expression of each community was aggregated to eigengenes, and the cause of death of GTEx donors was grouped based on the Hardy scale: (1) death by accident, blunt force trauma, or suicide-final stage estimated at <10 min; (2) rapid death from natural causes; (3) intermediate, i.e., patients who were ill but death was unexpected; (4) slow death after long illness. Differences in transcriptional expression according to cause of death were examined using one-way ANOVA. Differential expression of network communities in patients in relation to heart failure controls and propensity-matched controls from GTEx was analyzed by one-way ANOVA, after batch-effect removal of the sequencing data using combat, normalization, and aggregation into eigengenes.

### Statistics and reproducibility
Statistical tests used are described in the relevant sections, and nominal *P*-values below 0.05 were considered statistically significant unless stated otherwise. All analyses were performed using R programme (v4.1.0), and visual representation of co-expression network was generated using Gephi software (v0.9.2).

### Ethical approval
This study has been performed in accordance with the Declaration of Helsinki and approved by the Swedish National Council on Medical Ethics (reference number 2007/1410-31/3). Informed consent was obtained from all participants in both oral and written form. The original studies from which publicly available datasets have been used were all approved by their respective Institutional Review Boards.

## Results
### Clinical characterization of patients and controls
The detailed demographic and clinical characteristics of the participants are shown in Table 1. VO$_{2peak}$ median was 14.6 mL/kg/min (IQR 12.0–17.8 mL/kg/min), and time spent physically active (>100 counts/min) was 950 (IQR 541–1366) min/week. Of the patients monitored, 42 died during follow-up (median time: 1.8 years). All deaths were classified as cardiovascular events. In controls, VO$_{2peak}$ median was 25.4 mL/kg/min (IQR 21.9–29.4 mL/kg/min) and time spent physically active (>100 counts/min) was 2182 (IQR 1592–2790) min/week. The baseline characteristics of the heart failure patients included in this study were, with the exception of a higher-than-average use of device therapy and slight overrepresentation of males (81% vs 76%), not significantly different from all patients with HFrEF admitted to the Cardiology Department at Karolinska University Hospital during the study period, which deemed the current cohort representative of the target population (Supplementary Table 1). The initial control group recruited from patients admitted to the center on suspicion of heart failure but with normal echocardiograms and natriuretic peptides had a higher proportion of female subjects and a lower proportion of comorbidities (e.g., diabetes) compared to the heart failure group. To minimize the risk of differences in gene-expression between patients and controls being driven by such confounding factors, a secondary set of controls was generated through a balanced propensity-score matched (utilizing age, sex, BMI, and presence of diabetes) subset from the GTEx cohort. Baseline characteristics of the propensity-matched controls samples from GTEx cohort are outlined in Supplementary Table 2.

### Identification of a reproducible skeletal muscle transcriptome and construction of a heart failure co-expression network
In total 8867 genes passed the inclusion criteria considering all three datasets (heart failure cohort and confirmed expression in STRRIDE and GTEx). Of those, 852 were validated at the protein level in healthy skeletal muscle[23] (Human Protein Atlas - https://www.proteinatlas.org) A final network was constructed from an initial weighted gene co-expression network

constructed from the 58 heart failure patients by scale-free topology after minimizing the risk of spurious gene-gene correlations and after validation of the network communities. The final network consisted of 1481 transcripts with a total of >1 million validated gene-gene correlations divided into 14 different communities. A comprehensive list of the entire transcriptional network can be found in Supplementary Data 2.

## Biological characterization of network communities

Network communities were evaluated through gene ontology analysis to characterize each community in terms of biological functions and canonical pathways (Fig. 3). Five communities consisted of transcripts involved in various muscle metabolic processes: mitochondrial fatty acid (FA) beta-oxidation, glycolysis, oxidative phosphorylation, electron transport, and muscle contraction. Another group of 5 communities was enriched for processes related to cell adhesion, extracellular matrix (ECM) remodeling, and various immune responses. Three communities were enriched for various processes related to protein synthesis and protein turnover, and one community was enriched for cell cycle control and p53 signalling. A complete list of all significant enrichments for each community is provided in Supplementary Data 2. To further characterize the biology of the network communities, each community was also tested for enrichment of genes shown to be regulated in the skeletal muscle in other conditions of relevance: supervised exercise training[19], long-term bed rest (i.e., severe inactivity)[20], cancer cachexia[21], and cellular senescence[22]. The regulation of the different network communities in these clinical conditions is schematically shown in Fig. 3 and more comprehensively in Supplementary Data 2.

## Network community expression in relation to aged-comorbidity controls, clinical-derived variables, and prognosis

Compared to controls, the heart failure cohort showed higher expression of the p53 signalling (2.4-fold enrichment, $P = 3.05 \times 10^{-4}$) community and lower expression of communities related to mitochondrial FA beta-oxidation ($-2.1$-fold enrichment; $P = 0.002$), ECM remodeling ($-1.2$-fold enrichment; $P = 0.001$), oxidative phosphorylation ($-3.3$-fold-enrichment; $P = 1.22 \times 10^{-4}$), and muscle contraction ($-2.0$-fold enrichment; $P = 1.95 \times 10^{-5}$). These differences in gene-expression in relation to control participants were further corroborated by analysis of propensity-matched samples from the GTEx dataset: the p53 signalling community was up-regulated in heart failure patients compared with GTEx controls (3.0-fold enrichment; $P = 1.85 \times 10^{-16}$), while opposite trend, was observed for ECM remodeling community (1.1-fold enrichment; $P = 8.75 \times 10^{-9}$), and the mitochondrial FA beta-oxidation community (1.3-fold enrichment; $P = 0.007$).

By using eigengenes, the network communities were related to clinical characteristics of interest: NT-proBNP, LVEF, $VO_{2peak}$, daily physical activity, and prognosis. Nine communities were significantly correlated with at least 1 of the 5 clinically characteristics, and 3 communities (ECM remodeling, mitochondrial FA beta-oxidation, and p53 signalling) were correlated with ≥2 (Fig. 4, Supplementary Data 2). The 3 network communities with significant correlations with ≥2 clinical characteristics were further divided into tertials and tested for their relationship with selected clinical variables using one-way ANOVA (Fig. 5a-c). ECM remodeling and mitochondrial FA beta-oxidation followed a similar pattern; these communities had lower expression in patients compared with controls, lower expression in patients with low daily physical activity, and high NT-proBNP, respectively. An opposite pattern was observed for p53 signalling community with higher expression in patients than in controls. In addition, an increased expression was observed in patients with high NT-proBNP and more impaired systolic function. The prognostic potential of the network communities was assessed through Cox regression analysis (Fig. 6a-c). Significant associations with prognosis were found for p53 signalling ($P = 8.39 \times 10^{-4}$) and, to a lower extent, mitochondrial FA beta-oxidation community ($P = 0.026$). No significant association was observed for ECM remodelling and prognosis ($P = 0.209$). Similar associations were observed in the GTEx cohort where skeletal muscle expression of the p53 community

was elevated ($P < 2.00 \times 10^{-16}$), and mitochondrial FA beta-oxidation and ECM remodeling communities downregulated ($P < 2.00 \times 10^{-16}$ and $P = 0.014$, respectively) in patients succumbing to chronic illness (Fig. 7a-c). Finally, there was a significantly higher expression of p53 ($P = 0.017$) community observed in patients with a background of heart disease compared with age-matched patients with no prior record of heart disease in the GTEx cohort. The opposite trend was observed for FA beta-oxidation community ($P = 0.043$), with no significant difference in ECM remodeling community (Fig. 7d-f).

A closer look into individual genes showed that in the ECM remodeling community, a key component *COL1A2* (Collagen type I) was found to be downregulated 0.7-fold ($P = 0.001$) in patients compared to controls, but without significant association with prognosis, hazard ratio 0.7 ($P = 0.09$). In the mitochondrial FA beta-oxidation community, a key component was *MYORG* (Myogenesis Regulating Glycosidase), which was downregulated 0.6-fold ($P = 2.34 \times 10^{-6}$) in patients compared to controls and with a significant association with prognosis, hazard ratio 0.4 ($P = 4.11 \times 10^{-4}$) per copy number increase in heart failure patients. Finally, we identified a central regulator in the p53 signalling community, *CDKN1A* (p21). This factor was 1.8-fold up-regulated ($P = 0.003$) in relation to control-subjects and had a hazard-ratio of 1.4 ($P = 0.019$) per copy-number increase in gene-expression in the Cox-regression model. We also observed a significant dose-dependent increase in expression of *CDKN1A* in relation to NT-proBNP ($r = 0.42$, $P < 2.10 \times 10^{-4}$), and decreasing systolic function ($r = -0.34$, $P = 0.002$). A comprehensive summary of all transcripts in the network, their regulation in patients vs controls, and in relation to prognosis is provided in Supplementary Data 2.

## Discussion

In heart failure patients, dyspnea and fatigue[3–5] worsen in parallel with the severity of the disease and are among the most crucial factors in the deterioration of quality of life[6]. In addition, exercise capacity has repeatedly been shown to be a prognostic marker in heart failure[5,9] and the importance of skeletal muscle function for exercise capacity appears to be greater in heart failure patients than in healthy individuals[4,7,8]. Numerous studies have reported changes in selected proteins and transcripts in skeletal muscle of heart failure patients[24–27]. However, to our knowledge this is the first comprehensive transcriptional profiling and data-driven contribution to this field putting together pathways and possible skeletal muscle mechanisms with clinical characteristics and prognosis in patients with severe heart failure.

In this report, we constructed and validated a transcriptional network in skeletal muscle with 14 distinct communities involved in well-established biological processes in human skeletal muscle (Fig. 3). Compared to age-matched controls, we found differences at the transcriptional level that are consistent with the known skeletal muscle phenotype of heart failure[4,7,8]. These results confirm the validity of the current network-based analytical strategy (Fig. 2). To identify skeletal muscle processes of prognostic or pathophysiological significance, network communities were examined in relation to clinically relevant variables, $VO_{2peak}$, daily physical activity[5,9], NT-proBNP and LVEF[1], which describe different but distinct aspects of heart failure disease. The current results show significant associations between these factors and different transcriptional networks, but with different strength and number of associations (Fig. 4). Three network communities, related to ECM remodeling, mitochondrial FA beta-oxidation, and p53 signalling, were found to correlate with ≥2 clinical features, suggesting that these communities may be important signatures of skeletal muscle remodelling in heart failure.

The ECM remodeling community was downregulated in patients with heart failure compared with control subjects and its expression correlated with daily physical activity (Fig. 5a), but was not found significantly associated with prognosis (Fig. 6a). The observed correlation with daily physical activity was confirmed using gene-expression data from a supervised exercise intervention. In animal studies, acute and transient changes in ECM occur during the progression of heart failure disease, thus suggested to be

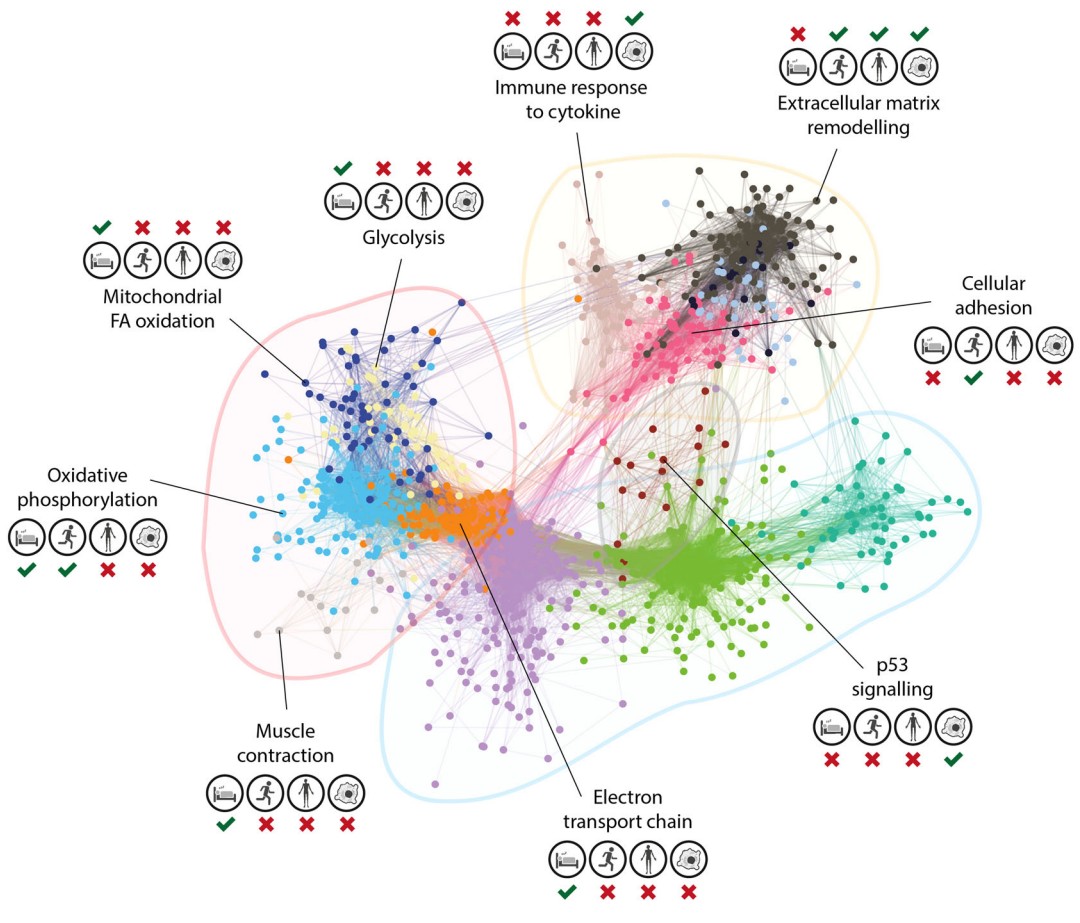

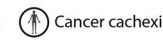 Bed-rest 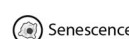 Exercise intervention · Cancer cachexia · Senescence

**Fig. 3 | Final transcriptional network and network characterization.** The final co-expression network derived from human skeletal muscle samples of patients with severe heart failure consisted of 1481 transcripts (dots). The network was divided into four branches with a total of 14 distinct communities (denoted by color) of transcripts with strong mutual correlations (lines) and with confirmed significant external validity in at least one of the two validation cohorts representing transcriptionally profiled skeletal muscle of >600 individuals. Each network community was characterized with regards to biological processes, canonical pathways and metabolites through enrichment analyses and the communities were labeled based on the most highly enriched term. The communities of the final network were thereafter tested for enrichment of genes established to be regulated in clinical conditions with relevance in the context of heart failure: Physical deconditioning through prolonged bed rest, regular exercise training, cancer cachexia and cellular senescence (denoted with icons). Created with icons from BioRender (2025; https://BioRender.com/4pw1fl6).

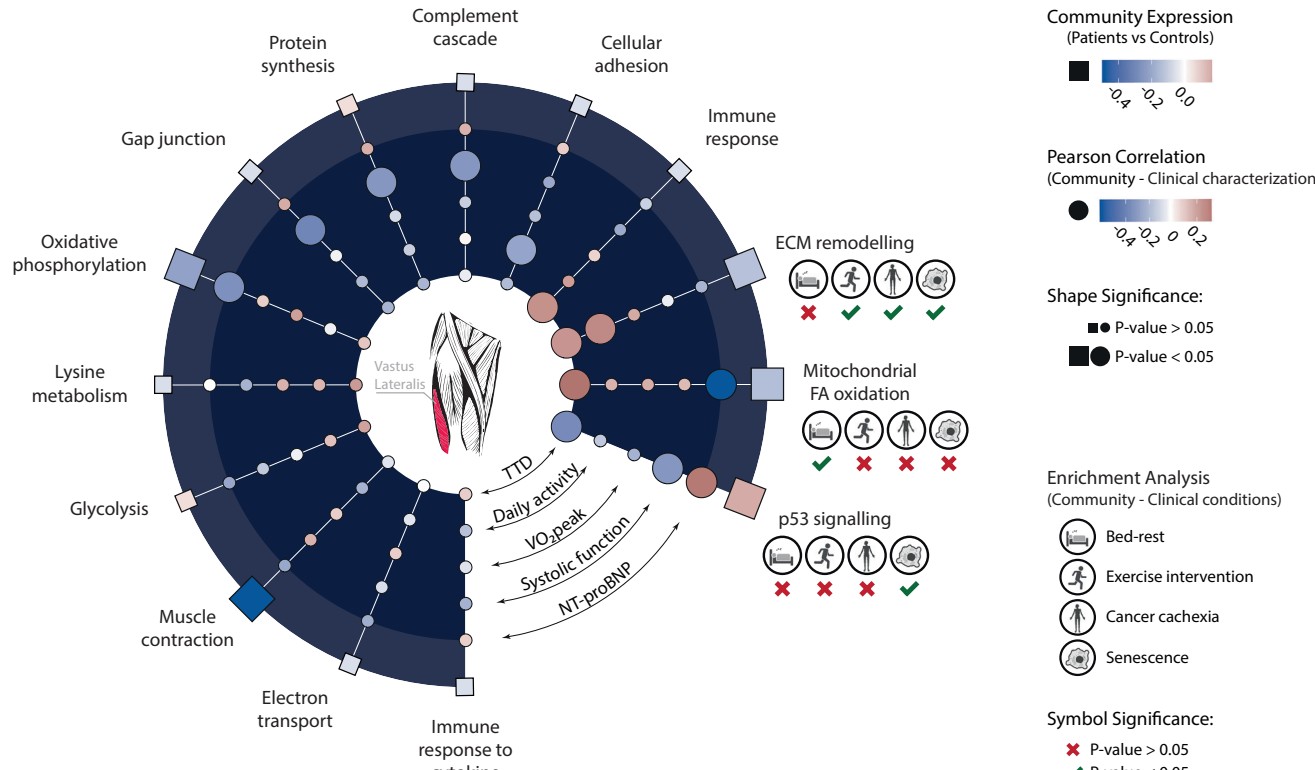

**Fig. 4 | Associations between network communities and clinical characteristics in patients with heart failure.** Community-level gene expression (eigengene) was compared between patients and controls using a two-sided Welch's two-sample t-test (outer ring, denoted by squares). Five communities differed significantly, of which all but one (p53-signalling) had lower expression in patients compared with controls. Interindividual variation in gene expression of the network communities were also correlated with clinical traits in heart failure using Pearson correlation analysis (inner ring, denoted by circles). Characteristics of interest were plasma NT-proBNP, cardiac systolic function in the form of LVEF, exercise capacity in the form of peak VO2, daily physical activity assessed though accelerometer measurements and time to death (TTD) in the form of time from baseline characterization to mortality event. Three communities, p53-signalling, mitochondrial FA beta-oxidation and ECM remodeling were both differentially regulated in patients compared to controls and showed significant associations with ≥2 clinical traits. Created with icons from BioRender (2025; https://BioRender.com/5sel492).

associated with disease progression in heart failure and of importance to skeletal muscle dysfunction[28]. The present findings support only the latter but not the former, as the dysregulation of ECM appears to be associated with general deconditioning. This is supported by the over-representation of the ECM community among genes found to be regulated in the cancer cachexia and CellAge datasets[21,22].

Expression of the network community involved in mitochondrial FA beta-oxidation was also downregulated in patients compared with controls, in a manner consistent with our analysis of the transcriptional signature of long-term bed rest in healthy individuals. Importantly, however, there was no significant association with daily physical activity in the heart failure cohort and no enrichment of genes found to be regulated by physical exercise (Fig. 5b). This suggests that this network community is not regulated by physical activity (or lack thereof) in the range of physical activity in normal life but rather is downregulated under more severe conditions such as heart failure or severe inactivity (i.e., long-term bed rest). In addition, the community was not enriched in genes from cancer cachexia transcriptional signatures, suggesting that the downregulation seen in heart failure is not a consequence of severe disease in general or negative energy balance. Moreover, compared with healthy controls, this community was downregulated, and negatively correlated with NT-proBNP and with prognosis (Fig. 4, Fig. 6b). Thus, lower gene expression in the mitochondrial FA beta-oxidation network was associated with higher NT-proBNP and poorer prognosis.

The observed downregulation of mitochondrial and ECM-related genes in patients is likely due, in large part, to their significantly lower daily physical activity levels, as measured by accelerometer data. This conclusion

is supported by the strong positive correlation we found between daily activity and the expression of these genes. A key finding was that the relationship between accelerometer-derived activity measures and gene expression was nearly identical in both patients and age-matched controls. This parallel suggests that transcriptional regulation of these pathways in response to habitual activity remains intact and does not appear to be a primary site of resistance to physical activity in this context. Therefore, while anabolic resistance may manifest in other ways, our data indicate that transcriptional differences in these specific pathways are more closely associated with differences in daily activity levels than with a fundamental difference in the muscle's gene regulatory response to physical activity.

The network community with the strongest association with prognosis was the p53 signalling community (Fig. 6c), confirming previous reports that highlight the importance of the p53 pathway in the pathogenesis of heart failure[29]. The p53 signalling community was enriched for genes associated with senescence but not with genes associated with supervised exercise intervention, bed rest, or cancer cachexia. The association of p53 signalling community with prognosis found in the current cohort was not a spurious correlation occurring only in the present study. Indeed, in the GTEx dataset, expression of this community was also increased in individuals who succumbed to chronic disease with heart disease compared with age-matched individuals (Fig. 7d), serving as an external validation of this finding.

The importance of the p53 signalling community was further underscored by the expression pattern of *CDKN1A* (p21). This important transcription factor exhibited higher expression in patients with heart failure and a hazard ratio of 1.4 per copy number increase in gene expression, estimated by the Cox regression model. Moreover, *CDKN1A* expression

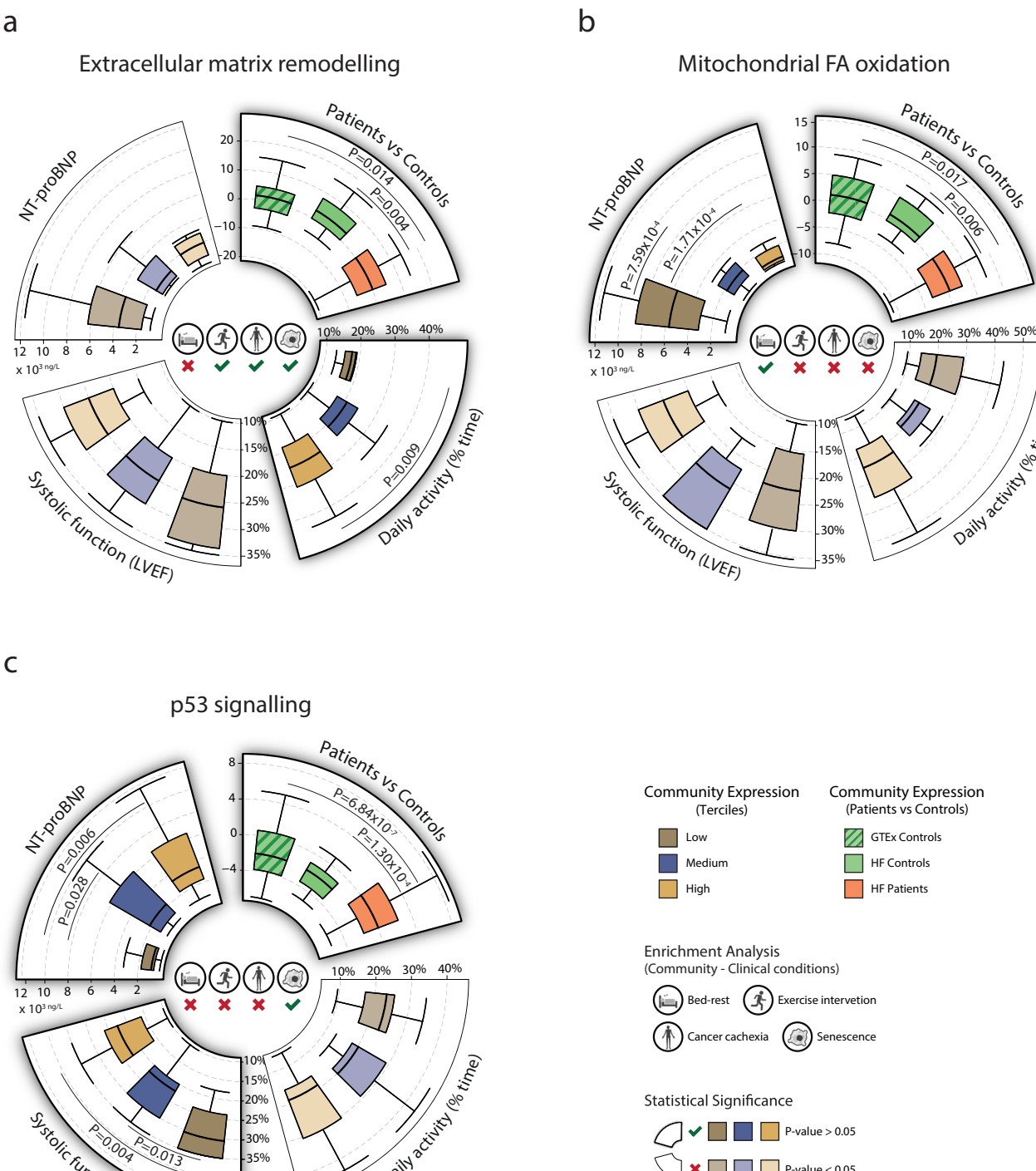

**Fig. 5 | Skeletal muscle gene expression in heart failure with regards to clinical traits.** In depth analysis of community-level gene expression (eigengene) in relation to clinical traits in heart failure cohort. **a** The ECM community was down-regulated in heart failure patients compared with controls ($P = 0.004$) and GTEx subjects ($P = 0.014$) and was positively associated with higher levels of daily physical activity. **b** The mitochondrial FA beta-oxidation community was down-regulated in patients compared with controls ($P = 0.006$) and GTEx subjects ($P = 0.017$) and was successively higher expression was associated with lower levels of NT-proBNP. **c** The

p53 signalling community had significantly higher ($P = 1.30 \times 10^{-4}$) transcriptional expression in heart failure patients compared with controls and in relation to propensity-matched subjects from the GTEx cohort ($P = 6.84 \times 10^{-7}$). Transcriptional expression was associated with lower LVEF and successively higher levels of NT-proBNP. All tests are two-sided Welch's two-sample t-tests. Created with icons from BioRender (2025; https://BioRender.com/4pw1fl6).

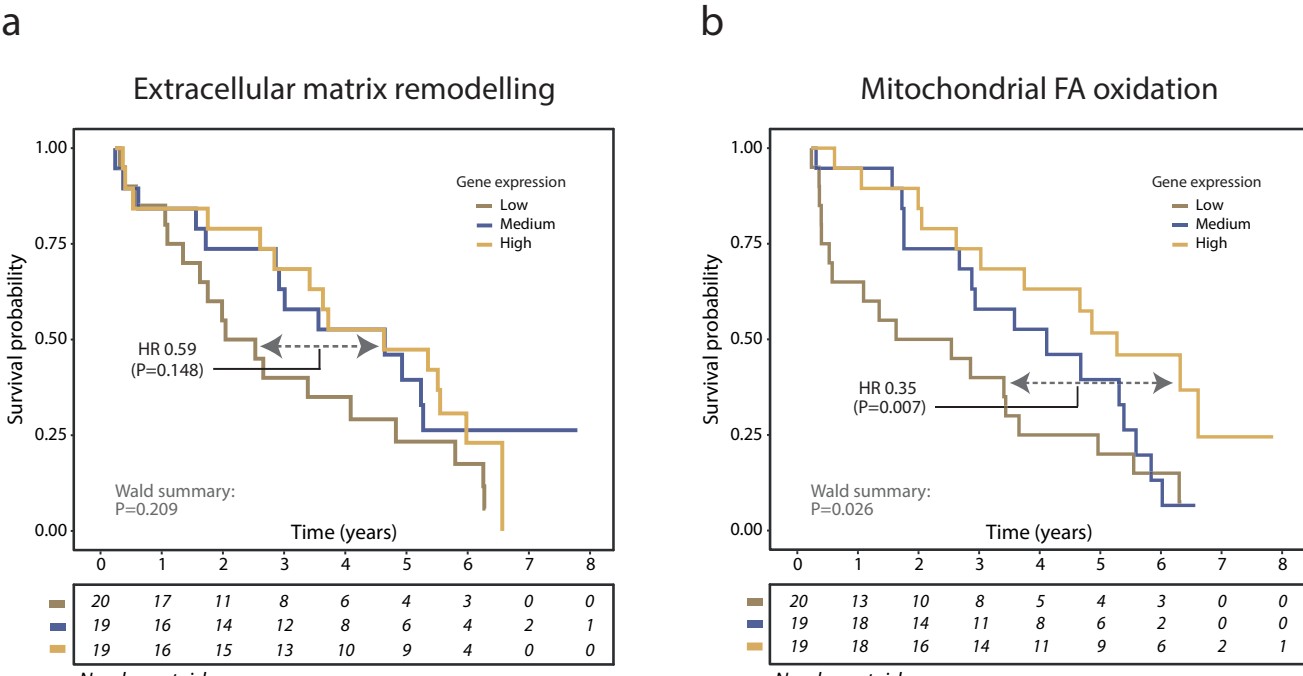

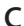

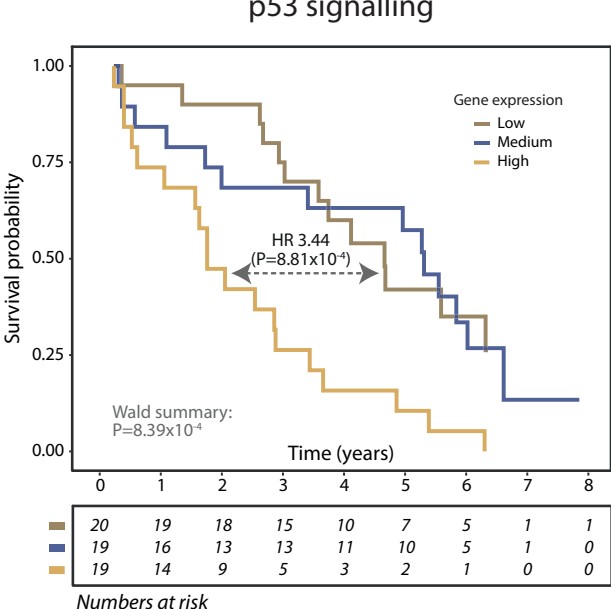

**Fig. 6 | Prognostic impact of skeletal muscle gene-expression in heart failure with regard to mortality.** Community-level gene expression (eigengene) was divided into terciles and assessed for prognostic value using Cox regression; **a** No significant association was observed for ECM remodelling and mortality (HR = 0.59, $P = 0.148$) however, higher expression was related to better prognosis. **b** In contrast, significant association with prognosis was observed for mitochondrial FA beta-oxidation community where higher expression was associated with better prognosis (HR = 0.35, $P = 0.007$). **c** Expression of the p53 signalling community was also associated with prognosis (HR = 3.44, $P = 8.81 \times 10^{-4}$) where higher gene-expression was associated with worsening prognosis. Wald tests were used to evaluate the Cox-models.

showed a dose-dependent increase in relation to NT-proBNP and decreasing systolic function. Upregulation of *CDKN1A* is a critical step in the conversion of proliferating cells to senescence[30] and thus represents a potential gatekeeping mechanism in terms of deteriorating muscle function and prognosis. The plausible relevance of the current finding in the development of heart failure skeletal muscle dysfunction is that (1) the accumulation of *CDKN1A*-positive skeletal muscle nuclei is a characteristic feature of frailty in an elderly population[31]. (2) Activation of the p53 signalling pathway has been shown to be a feature of cellular senescence in myocytes and biologically aged skeletal muscle fibers, as it is associated

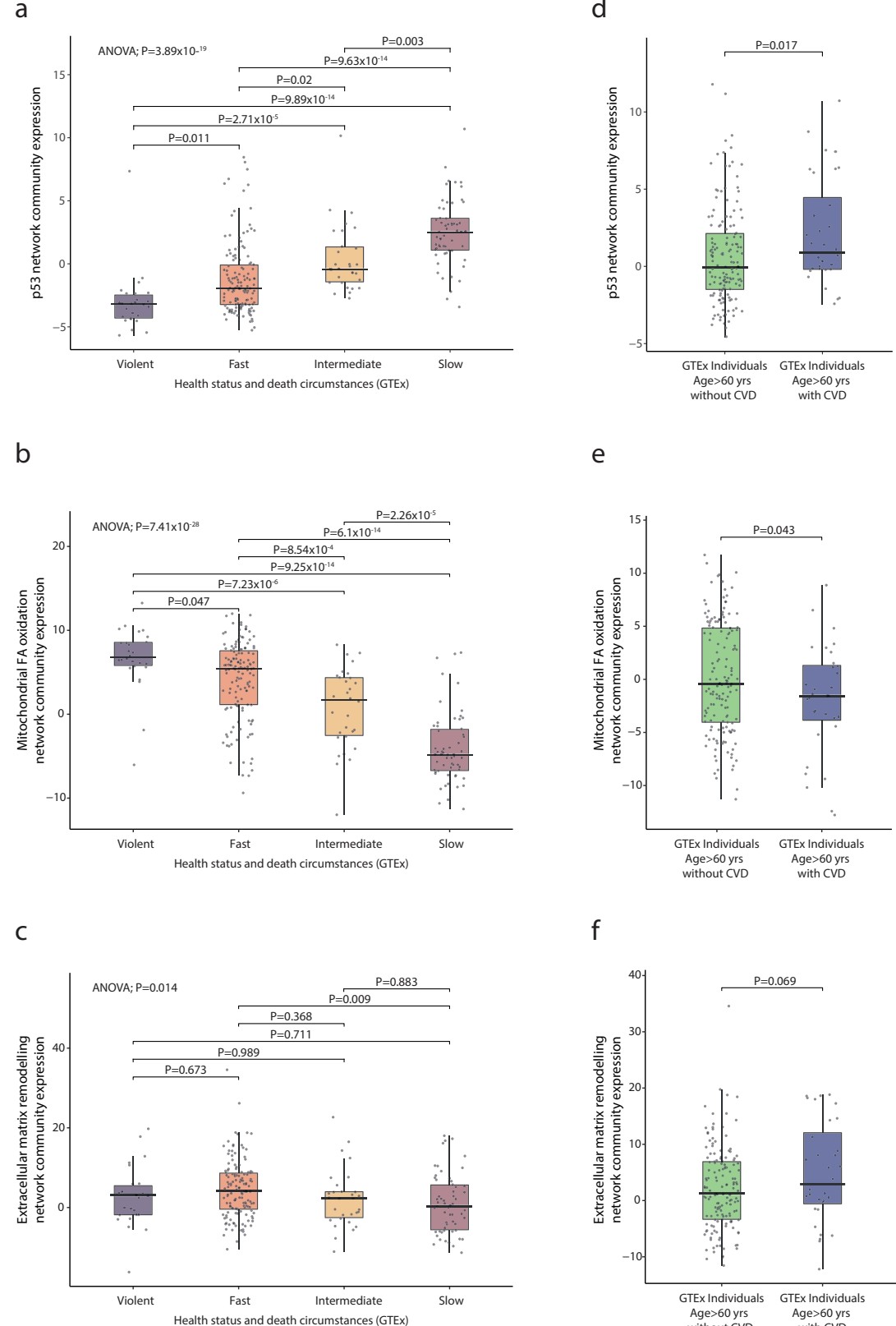

**Fig. 7 | External validation of the prognostic value of skeletal muscle gene-expression in relation to cause of death. a–c** Gene-expression of the p53 network community in relation to health status and circumstances of death: The GTEx dataset provides information the prevalence of cardiac disease and the time foregoing death on all samples so-called hardy-scale which divides patients into categories as follows; Sudden unexpected death where subject can be assumed to be previously healthy (Violent). Sudden unexpected deaths of people who had been healthy (Fast), after a terminal phase estimated at <1 hr. Death after a terminal phase of 1 to 24 hrs, patients who were ill, but death was unexpected (Intermediate). Death after prolonged illness, with a terminal phase longer than 1 day i.e., deaths that are not unexpected (Slow). **d–f** Skeletal muscle gene-expression of network communities in patients >60 years of age with or without established cardiovascular disease prior to death in the GTEx cohort ($n = 252$). Statistical inference was done using one-way ANOVA.

with the regenerative capacity of myofibers[32]. Taken together, this suggests that altered cell cycle control, i.e., cellular senescence, at the skeletal muscle level is involved in the pathophysiology of heart failure and represents a prognostically relevant process that is not primarily influenced by physical inactivity, general deconditioning, or negative energy balance[32,33].

Understanding the interplay between muscle function and disease prognosis not only contributes to our understanding of disease mechanisms but also has significant clinical implications. The observation that gene communities regulated by variations in physical activity show a weaker association with prognosis than other disease-specific gene clusters highlights the potential of muscle dysfunction as a biomarker of disease severity and progression. If incorporated into clinical assessment, muscle dysfunction could improve prognostic accuracy, help identify high-risk patients and guide personalized treatment strategies. Furthermore, the contrast between the p53-associated gene community, which is not affected by exercise-related genes, and the ECM remodeling and mitochondrial FA beta-oxidation community underscores the importance of distinguishing between disease-specific dysregulation and disruptive effects of physical (in)activity. This distinction is important for identifying the molecular pathways responsible for disease development and thus finding promising targets for therapeutic intervention.

This study has its limitations. Most importantly, it is an observational study that requires cautious interpretation of the results in terms of pathophysiologic mechanisms and predictions, but we cannot establish causality due to potential confounding variables and biases. To strengthen the validity of our observations we used internal (i.e., correlations within the patient group) and external (i.e., confirmation of network communities in external datasets) validation, both with regard to clinical data and gene-expression. However, further experimental and translational studies are needed to investigate whether the transcriptional communities presented here are actually influencing the pathophysiology of heart failure or are merely a consequence of disease severity. Although the studied group is relatively large for molecular characterization/RNA sequencing, it is still a very small sample in comparison to clinical or epidemiological studies, and the effects of several potentially relevant strata, such as age, gender, comorbidity, and differences in medical therapy, on skeletal muscle in heart failure are therefore not investigated in the present study.

The control group consisted of patients who were referred for a clinical examination with echocardiography and analysis of natriuretic peptides due to dyspnea, however natriuretic peptides and echocardiogram showed no evidence of heart failure. This resulted in the control group being comparable to the patient group in terms of daily physical activity and comorbidities. However, this also introduced a bias, as the control group was slightly older and with higher proportion of female participants compared to the patient group. This was mitigated by adding an age- and sex-matched control sample from the GTEx dataset and by analyzing the correlation between gene expression and clinical characteristics only within the patient group. The association between gene expression and prognosis observed in the current study was externally validated in relation to mode of death in the GTEx dataset rather than in an external heart-failure dataset, as no such dataset is publicly available. The validation analysis carried out in GTEx dataset may therefore suggest that the results are general rather than specific to patients with severe heart failure. Finally, the findings presented here are based on the integrated analysis of several publicly available data sets. Despite this being a common practice, such integration oftentimes introduces technical artefacts that can be to an extent mitigated but not fully eliminated. As such, this remains a challenge of the field and a limitation of the current setup.

## Conclusion
Using co-expression analysis, we identified several network communities in skeletal muscle of patients with severe heart failure that showed altered expression levels compared to control subjects. Among these, communities associated with processes linked the skeletal muscle hypothesis (e.g., ECM signalling or mitochondrial FA beta-oxidation communities) were also strongly enriched for genes regulated by variations in physical activity, i.e.,

bed rest and exercise, suggesting that their dysregulation is, to a large extent, driven by inactivity in patients with heart failure. Interestingly, these communities demonstrated a weak association with prognosis compared with the p53 signalling community, which was not associated with variations in physical activity. This suggests the importance of the p53 signalling community in disease-specific dysregulation of skeletal muscle. Considering the association of the p53 signalling community with prognosis in the heart failure cohort was also validated in an external dataset serves as compelling evidence for the pathophysiological significance and clinical potential of this pathway.

Finally, our results underscore the importance of distinguishing between disease-specific dysregulation and negative effects of physical inactivity. This distinction is crucial for identifying the molecular pathways responsible for disease development and finding promising targets for therapeutic intervention.

## Data availability
The source data underlying Figs. 3, 4, 5a-c, 6a-c are found in Supplementary Data 3-5. The source data underlying Table 1 and Supplementary Table 1 are found in Supplementary Data 6 Heart failure RNA sequencing data is deposited at Gene-Expression Omnibus (GEO) (GSE262824; https://www.ncbi.nlm.nih.gov/geo/query/acc.cgi?&acc=GSE262824). RNA sequencing data of GTEx validation cohort is available at https://gtexportal.org/home/, and complete microarray gene-expression data of STRRIDE validation cohort is publicly available at GEO (GSE1295; https://www.ncbi.nlm.nih.gov/geo/query/acc.cgi?acc=GSE1295).

## Code availability
Code used for the generation of the co-expression network, performed statistical analysis, and generated figures can be shared upon request to the corresponding author.

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

## Acknowledgements
This work has been supported by grants from Swedish Research Council 2015-02338, Marcus and Marianne Wallenberg Foundation, Stockholm County Council, Swedish Society of Medical Research and Swedish Heart Lung Foundation (20220725).

## Author contributions
E.R., A.L., M.M., R.F.G., and T.G. designed the study and wrote the manuscript. M.M., E.R., and R.F.G. scheduled human subjects and performed human muscle biopsies. E.R. and A.L. performed all data analysis described in the manuscript. All authors have read and approved the final manuscript.

## Funding

## Competing interests
The authors declare no competing interests.
