## [Transparent Peer Review file · Communications Medicine]

Skeletal muscle transcriptional dysregulation of genes involved in senescence is associated with prognosis in severe heart failure

Corresponding Author: Dr Eric Rullman

Version 0:

Reviewer comments:

Reviewer #1

(Remarks to the Author)

The present work is a comprehensive analysis of limb muscle transcriptomics in patients with severe heart failure with reduced ejection fraction and identifies gene pathways and networks that are modified in patients, correlate with clinical variables, and can distinguish outcomes. The data are novel and establish a strong set of genes that can be tested for causality. The statistical analysis are appropriate and extensive. I have some concerns regarding the conclusion and interpretations presented that are not supported by the data. My concerns are outlined below.

The points below are relevant throughout the manuscript (abstract, discussion, conclusion)

Line 1287-288: "...to identify genes prognostic or pathophysiological relevance..." the data does not really suggest any of that. It is a common misconception that cross-sectional observation with different response for mortality or outcomes has prognostic or pathophysiological relevance, and authors need to be careful.

Line 321-322...suggesting clinical relevance. Same as above. The observations are interesting and could be insightful suggesting pathways worth targeting to define clinical or pathological relevance but do not distinguish disease-causing vs disease-inducing effects.

Abstract/Conclusion: The authors must be careful to suggest that pathways overlapping in HFrEF and physical inactivity indicate the change is due to inactivity. Some of the correlations and data suggest inactivity is not a determining factor in several observations reported. There are alternative interpretations, e.g., the genes/pathways responsive to activity levels become more sensitive in HFrEF requiring higher than normal activity for repression/stimulation. Attributing the changes to inactivity suggests that enhancing physical activity per se is sufficient to solve the skeletal muscle problems in HFrEF. Although it brings benefits, there is no evidence it solves the problem. Conclusive evidence would come from comparing healthy controls post-training and HFrEF post-training/enhanced physical activity. Examining data from HFrEF post-training vs sedentary controls, which has been done, does not prove that enhanced activity solves the problem. It appears more likely that, similar to 'anabolic resistance' in healthy aging, patients with HFrEF have enhanced sensitivity to inactivity and/or activity-resistance.

Reviewer #3

(Remarks to the Author)

Rullman et al. performed a transcriptomic and bioinformatic analysis of human skeletal muscle biopsies from patients with HFrEF vs. non-failing controls. The authors performed additional validation and quality control of their RNA-seq data using several larger open-access datasets of human skeletal muscle transcriptomics. Network analysis revealed 14 unique gene-pathway communities, of which 5 (p53 signaling, mitochondrial FA oxidation, ECM remodeling, oxidative phosphorylation, and muscle contraction) showed differential enrichment in HF patients vs. controls. Correlation analyses revealed associations of 3 communities (ECM remodeling, mitochondrial FA oxidation, and p53 signaling) with clinical variables of systolic function, NT-proBNP, daily activity, and prognosis/death. These findings were further validated using age-, sex-, and

comorbidity-matched patients from the GTEx open access dataset. Correlation with other published datasets investigating muscle transcriptomic signatures with exercise, bed rest, cancer cachexia, and cell senescence revealed overlap between ECM remodeling pathways and gene changes seen in exercise, cachexia, and senescence, mitochondrial FA oxidation with inactivity/bed rest, and p53 signaling uniquely correlating with cellular senescence.

This is a detailed network level analysis involving a large number of human patient samples, which shows interesting insight into the molecular mechanisms underlying skeletal muscle dysfunction in the context of advanced heart failure, and the relative contribution of inactivity, general severe chronic illness, or heart failure-specific signals to the skeletal muscle and the association with HF prognosis.

Major comments:

- 1) There seems to be a discrepancy in the manuscript/figures regarding the number of patients included in the study. The text indicates 66 HF patients and 28 control participants were included in the study, whereas Fig 1 and Table 1 indicate 58 HF patients and 20 controls. This discrepancy needs to be addressed.
- 2) A significant finding of this study is the analysis of shared gene expression signatures from the skeletal muscle of HF patients vs. skeletal muscle after exercise, bed rest, cachexia, and cellular senescence, however, relatively little information is provided regarding these comparison datasets and enrichment analyses. For example, the size (# of genes) of the differential gene expression signatures pulled from these other studies and used for comparison should be included. Moreover, the authors indicate that a comprehensive analysis of the HF network communities in these clinical conditions is included in Additional file 3 (Line 221), however this data does not appear to be included anywhere in the manuscript (e.g. the Fisher exact enrichment level, p-values, etc.). These data should be included.
- 3) While the authors use a co-expression network analysis approach to interpret their transcriptomic findings, the manuscript would benefit from additional analyses or reporting of differential gene expression results. E.g. how many genes pass fold change cutoffs and FDR cut offs when comparing expression in HF vs. control? How many genes in each community also pass these cutoffs? What proportion of the differential HF skeletal muscle transcriptome is represented in each community?
- 4) There doesn't appear to be a rationale for the GTEx validation analysis in Fig 6 to not include the FA oxidation and ECM remodeling data. Fig 6 shows 2 separate analyses, i) correlation with expression and death from acute vs. slow/chronic cause, and ii) expression in individuals with and without CVD. It is unclear from the text what the indicated P-values are referring to without the associated figure.
- 5) How did the authors identify the single genes highlighted as key components of the individual communities? Are they ranked based on fold change in expression? (doesn't appear to be the case for COL1A2). In addition, the expression p-values in paragraph line 258-270 don't appear to match those provided in Additional file 3. For example, CDKN1A does not pass the FDR (0.07) and survival analysis p-value is indicated as 0.02 rather than <0.001. Illustrating the magnitude of these gene changes in relation to the rest of the dataset, e.g. highlighting these 3 genes on a volcano plot, would better highlight their importance as differential gene signatures in HF vs. control. Specific data (correlation strength, p-values) for CDKN1A correlation to NT-proBNP and LVEF should also be included in the manuscript if discussed in the results (line 266-268).
- 6) The authors use very creative visual representations of the bioinformatics data in the manuscript figures, however these figures (e.g. Fig 4, 5) aren't immediately intuitive to the reader. More details could be provided in the figure legends on the colour/symbol schemes and analyses shown in these graphics. An additional table providing the individual correlation data for each community and clinical variable would also benefit.

Minor comments:

- 1) Line 189 should indicate that the control group had a higher proportion of female subjects, not male.
- 2) In figure 3, the authors use a heart schematic as the central illustration. Given that this study is an analysis of skeletal muscle, a different illustration might be more appropriate to convey the study question.
- 3) Figure 3, the units for "DEG regulation, patients vs. controls" scale bar should be included, e.g. logFC.
- 4) Figure 6: panels A and B should be called out separately in the results.
- 5) Throughout the manuscript, when comparing the disease (HF) cohort to controls, it may be more accurate to describe any changes as they relate to the disease group, rather than the control group. E.g. Line 233 "ECM remodeling community was higher in GTEx controls" should be changed to "was lower in HF patients".
- 6) Paragraph, Line 236-244: the 4 clinically defined variables used in these comparisons are listed as VO2 peak, daily physical activity, NT-proBNP, and LVEF. From these variables, only 1 network community (p53) has significant correlations with more than 1 clinical variable, not 3 (+ ECM and FA oxidation), as written in the manuscript. When including death/prognosis as a 5th factor, then all 3 communities have 2 or more associations. How this is written, whether "death" is included or distinct from this list of clinical variables, should be revised.
- 7) Line 246: "respectively" should be added to this sentence as low physical activity was only true for ECM, and NT-proBNP for FA oxidation.
- 8) Line 248: p53 was not associated with daily physical activity in Fig 5, this sentence should be revised.
- 9) Line 296: sentence relates to data in Fig. 4 rather than Fig 3.
- 10) Line 342: This statement should be supported with references to the appropriate primary research articles.

Reviewer #4

(Remarks to the Author)

Rullman et al. present RNA-seq data from human skeletal muscle tissue in patients with HF vs. a variety of control populations. They perform a comprehensive network analysis and identify several gene networks that are dysregulated in association with HF. The study is novel in its use of RNA-seq in skeletal muscle in this population, an organ commonly compromised but little studied. However, the study is limited as it is essentially a 1-experiment analysis without follow-up

confirmatory testing or experimental modeling of any of the identified networks. Overall, while interesting and rather novel, the manuscript suffers from poor organization and flow, and challenges associated with the depiction of the data.

Major Comments:

1. Figures and tables are hard to follow as they extend from p28-34 without any listing on those pages for what figure/table is what. Can a label be added to the page (I'm sure this could be deleted in the proofs) facilitate easier review? Similarly, the Additional files are not labeled as such in the files available in the reviewer folder. If these files are improperly titled, perhaps change the tab names for each file to ensure the reviewer knows which additional file is which. Also, the sample sizes for the groups referred to in the text for Additional files 1 and 2 is not stated in the text; this would have also made it easier to identify which file was which. I eventually figured it out but spent more time than was necessary doing so.
2. Fig 3 – the images in the circles need to be enlarged; they are rather hard to discern. Also, why are some black and some white? Is this representative of some other kind of difference? If just for contrast, please make them uniform as doing it this way makes it seem like there is a difference (which isn't defined in the legend). More importantly, are these icons validated or the author's self-characterization of the roles of these pathways (e.g., did patients actually meet the clinical definition of cachexia)? What are the checks and x's for? This is not well-described.
3. I'm sorry, but I simply do not understand what I'm looking at in Figure 5. There appear to be 3 sets of colors but only 2 are defined in the legend...2 of the 3 color schemes are relatively similar; are these supposed to be the same? If so, please make them uniform. Further, while the graphs used are clever and visually interesting, the data in them is too small to read, particularly the statistical comparisons. These need to be enlarged.
4. Line 238-9 indicates 9 gene communities were significantly correlated with 1 of the 4 clinical variables they studied. As best I can tell, they only show data on the 3 that were correlated with more than 1. These other data should also be presented, even if just in the data supplement.
5. First paragraph of discussion, they indicate their data as "describing the mechanisms" but this study is associative not causal (i.e., not definitively showing a mechanism, but rather shows an association between skeletal muscle gene networks and heart failure). They should rephrase this characterization.
6. When discussing beta-oxidation, they state, "in a manner consistent with our analysis of the transcriptional signature of long-term bed rest" Is there a study of the transcriptional signature in patients on long-term bed rest? If so, this needs to be cited/referenced. This is partly related to my question about the icons (see #2 above). I'm also confused by the frequent references to bed rest...I'm a HF physician, other than my sickest patients who are hospitalized, HF patients aren't on strict bed rest.
7. The flow of the discussion is a bit abnormal; they spend considerable time on the other pathways in the first half (more than half) of the discussion, only coming to the strongest signal and discussion point (p53 signaling in skeletal muscle) last. I would recommend a significant revision of the flow of this section.
8. Lastly, there is absolutely no discussion of study limitations, of which there are many.

Minor Comments

1. There are errors of formatting in the references (e.g., ref 4, "Working Group 'Exercise 400 Physiology SC and CR of the IS of C"; ref 10, "Haehling S von" etc...) that should be cleaned up.
2. Line 47 "HRrEF"
3. There are scattered grammatical and typographical errors.

Version 1:

Reviewer comments:

Reviewer #1

(Remarks to the Author)

The authors have dismissed my comments and provided a rationale for dismissal that is founded on one citation for a study showing that patients respond to training. The citation and the rationale do not address the issue of a blunted response, which is implied that activity resistance like anabolic resistance. The discussion is centered on the notion of muscles having an inactivity genetic footprint that the authors conclude results from inactivity. However, there are no reliable (quantitative) measurements of daily physical activity in the study, which would be required to support their stance.

Reviewer #2

(Remarks to the Author)

That authors have adequately addressed my previous concerns.

Remaining are a few minor points for revision:

- 1) It is difficult to correctly associate the various tabs provided in the additional data files to the appropriate section of the text. It would be helpful if the tabs could be numbered or labeled in a way that they could each be called out at the appropriate place in the manuscript text.
- 2) Figure 4 – The results of enrichment analysis of the gene communities to other clinical conditions (e.g. bed rest, exercise, cachexia, senescence) is provided in Figure 3 and does not necessarily need to be repeated in Figure 4.
- 3) Discussion Line 364: "suggesting that these communities are plausible key processes in the pathophysiology of skeletal

muscle in heart failure". This statement should be revised. That these "may be important signatures of skeletal muscle transcriptional remodeling in HF" would be more appropriate, as there is no data demonstrating an intrinsic functional decline in the skeletal muscle of these patients in the present study that can be correlated with these signatures.

Reviewer #3

(Remarks to the Author)

This is a revised manuscript. The authors have addressed my biggest concern regarding flow and readability, and the manuscript is far more readable now. Of note, they did not (as indicated in the rebuttal letter) bring the p53 story to the forefront of the discussion. However, the extensive changes make the flow substantially better, obviating the need for that revision. I have the following comments that should be addressed, but represent rather minor changes that can be addressed quickly without further experimental testing/analysis.

Major Comments:

1) I have 2 comments regarding the new limitations section. (1) in the first paragraph, they indicate that their design makes the validity "likely high" – this is subjective and biased wording and isn't definitively proven by the data in the manuscript. I'd like to see this sentence tempered to something more like, "We used ...internal/external...to improve/strengthen the validity of our observations." (2) One last but important limitation is that the other cohorts used for comparison would have had sequencing by different methodologies/different facilities. This makes them suboptimal controls – I totally understand why they were used (this is very hard tissue to obtain), but this needs to be acknowledged as a source of potential/probable bias.

Minor comments:

1. IQR is not defined at first use. I presume it means interquartile range (and that use is appropriate in the first section of the results. However, the first use is in the methods pertaining to the STRRIDE genes and I do not understand what they mean by identifying genes with IQR >2. Is this a typo in this spot?
2. For the new Additional Table 1, I suggest updating the "Heart Failure (N=58)" column header to something like "Study Patients (N=58)" as this would be clearer.

Version 2:

Reviewer comments:

Reviewer #2

(Remarks to the Author)

The authors have addressed my previous concerns.

Reviewer #4

(Remarks to the Author)

This is a 2nd revision of this manuscript. The authors have adequately addressed my concerns. I have no further substantive comments.

Reviewer #1 (Remarks to the Author):

The present work is a comprehensive analysis of limb muscle transcriptomics in patients with severe heart failure with reduced ejection fraction and identifies gene pathways and networks that are modified in patients, correlate with clinical variables, and can distinguish outcomes. The data are novel and establish a strong set of genes that can be tested for causality. The statistical analyses are appropriate and extensive. I have some concerns regarding the conclusion and interpretations presented that are not supported by the data. My concerns are outlined below.

The points below are relevant throughout the manuscript (abstract, discussion, conclusion).

Major comments:

1) Line 287-288: "...to identify genes prognostic or pathophysiological relevance..." the data does not really suggest any of that. It is a common misconception that cross-sectional observation with different response for mortality or outcomes has prognostic or pathophysiological relevance, and authors need to be careful.

REPLY: Thank you for your comment.

It is very important what kind of data can define/determine what, i.e. A) the driving processes in the disease contain information about prognostic value, B) pathophysiologic processes both secondary effects and specific processes (see A), C) contribute to functional status (such as quality of life, physical status, etc.) and D) contain information about prognostic value. As we understand the comment, the reviewer's main concern is that with the current study design we can define what drives the disease process. In this paper we have tried to answer the question raised in various ways.

A simple comparison of gene expression between the control group and the group with heart failure should end up in the problem you mentioned. Here, we have tried to develop a strategy that allows us to better explain the observed difference to the corresponding control group, at least to a small extent, and also to evaluate the influence on the disease process. In the current study, we first identified networks that differed from the control group and then in the HF cohort linked the differential expression to suggested driving factors in heart failure progression (such as cardiac function, ANP, etc.). The network associated with ≥ 2 of these factors was selected. To get a better picture of the mechanisms underlying the changes in gene expression in these networks, we analyzed gene expression in these networks with respect to known exogenous factors associated with heart failure that are known to influence gene expression. Therefore, we used data describing gene changes in skeletal muscle during different types of physical activity, unloading/bed rest, cachexia and aging. Importantly, for physical activity, the measured changes in gene expression in the network were also related to physical activity in the accelerometer-measured cohort.

Finally, these networks were also prospectively associated with clinical outcome. Thus, we believe that this approach and the clarity of its description, together with the way how we express ourselves in the revised paper, minimize the risk of a false impression of the study-design.

2) Line 321-322...suggesting clinical relevance. Same as above. The observations are interesting and could be insightful suggesting pathways worth targeting to define clinical

or pathological relevance but do not distinguish disease-causing vs disease-inducing effects.

REPLY: Thank you for your comment. See reply above.

3) Abstract/Conclusion: The authors must be careful to suggest that pathways overlapping in HFrEF and physical inactivity indicate the change is due to inactivity. Some of the correlations and data suggest inactivity is not a determining factor in several observations reported. There are alternative interpretations, e.g., the genes/pathways responsive to activity levels become more sensitive in HFrEF requiring higher than normal activity for repression/stimulation. Attributing the changes to inactivity suggests that enhancing physical activity per se is sufficient to solve the skeletal muscle problems in HFrEF. Although it brings benefits, there is no evidence it solves the problem. Conclusive evidence would come from comparing healthy controls post-training and HFrEF post-training/enhanced physical activity. Examining data from HFrEF post-training vs sedentary controls, which has been done, does not prove that enhanced activity solves the problem. It appears more likely that, similar to 'anabolic resistance' in healthy aging, patients with HFrEF have enhanced sensitivity to inactivity and/or activity-resistance.

REPLY: Thank you for your comment.

We understand the reviewer's concerns, but would also like to point out that the study includes an age-matched reference group (with similar age induced anabolic resistance) and that accelerated anabolic resistance in HF is not supported by several publications showing peripheral skeletal muscle adapts quite rapidly to exercise in patients with chronic heart failure (Magnusson et al. Eur Heart J 1996 Jul;17(7):1048-55.).

In current study, genes defined as influenced by physical activity were based on a combination of published gene expression profiles after exercise but also had to be expressed in a dose-dependent fashions in relation to accelerometer-measured physical activity in the HF-patients. Therefore, we would argue that the risk of a HF-specific anabolic or aerobic resistance to be the underlying mechanism is unlikely.

Reviewer #3 (Remarks to the Author):

Rullman et al. performed a transcriptomic and bioinformatic analysis of human skeletal muscle biopsies from patients with HFrEF vs. non-failing controls. The authors performed additional validation and quality control of their RNA-seq data using several larger open-access datasets of human skeletal muscle transcriptomics. Network analysis revealed 14 unique gene-pathway communities, of which 5 (p53 signaling, mitochondrial FA oxidation, ECM remodeling, oxidative phosphorylation, and muscle contraction) showed differential enrichment in HF patients vs. controls. Correlation analyses revealed associations of 3 communities (ECM remodeling, mitochondrial FA oxidation, and p53 signaling) with clinical variables of systolic function, NT-proBNP, daily activity, and prognosis/death. These findings were further validated using age-, sex-, and comorbidity-matched patients from the GTEx open access dataset. Correlation with other published datasets investigating muscle transcriptomic signatures with exercise, bed rest, cancer cachexia, and cell senescence revealed overlap between ECM remodeling pathways and gene changes seen in exercise, cachexia, and senescence, mitochondrial FA oxidation with inactivity/bed rest, and p53 signaling uniquely correlating with cellular senescence.

This is a detailed network level analysis involving a large number of human patient samples, which shows interesting insight into the molecular mechanisms underlying skeletal muscle dysfunction in the context of advanced heart failure, and the relative contribution of inactivity, general severe chronic illness, or heart failure-specific signals to the skeletal muscle and the association with HF prognosis.

Major comments:

1) There seems to be a discrepancy in the manuscript/figures regarding the number of patients included in the study. The text indicates 66 HF patients, and 28 control participants were included in the study, whereas Fig 1 and Table 1 indicate 58 HF patients and 20 controls. This discrepancy needs to be addressed.

Reply: Thank you for your comment. We apologize for any inconvenience this may have caused. This has now been corrected in the revised version.

2) A significant finding of this study is the analysis of shared gene expression signatures from the skeletal muscle of HF patients vs. skeletal muscle after exercise, bed rest, cachexia, and cellular senescence, however, relatively little information is provided regarding these comparison datasets and enrichment analyses. For example, the size (# of genes) of the differential gene expression signatures pulled from these other studies and used for comparison should be included. Moreover, the authors indicate that a comprehensive analysis of the HF network communities in these clinical conditions is included in Additional file 3 (Line 221), however this data does not appear to be included anywhere in the manuscript (e.g. the Fisher exact enrichment level, p-values, etc.). These data should be included.

Reply: Thank you for the comment. The data and information requested by the reviewer has now been added Additional file 2 (former Additional file 3), under following sheet names: Community info and validation; Gene info and analysis.

This includes:

- ***Results of over-representation analysis between gene communities and genes deemed significantly differentially expressed in above mentioned datasets.***

We have now amended the methods-section with a more detailed and precise description of the four datasets and how the gene-sets were defined. This is accompanied with differential expression results from original publications.

3) While the authors use a co-expression network analysis approach to interpret their transcriptomic findings, the manuscript would benefit from additional analyses or reporting of differential gene expression results. E.g. how many genes pass fold change cutoffs and FDR cut offs when comparing expression in HF vs. control? How many genes in each community also pass these cutoffs? What proportion of the differential HF skeletal muscle transcriptome is represented in each community?

Reply: Thank you for the comment. While we agree with the reviewers' comment we chose not to overwhelm the main text and the reader with extensive data. Rather we provided this information in Additional file 2 however, in somewhat different form.

This includes:

- **Result of differential expression analysis between HF patients and controls for all genes in the network.**
- **Instead of reporting numbers based on the arbitrary cutoffs and overlaps we also included results of over-representation analysis between gene communities and genes deemed significantly differentially expressed between HF patients and controls. This includes P-values from Fisher exact test and corresponding percentage of significant genes present in each community. We believe this to summarize the above suggestions within one metric.**

We also want to emphasize transparency of data/results provided in Additional file 2 allowing the reader to independently calculate requested statistics if needed.

4) There doesn't appear to be a rationale for the GTEx validation analysis in Fig 6 to not include the FA oxidation and ECM remodeling data. Fig 6 shows 2 separate analyses, i) correlation with expression and death from acute vs. slow/chronic cause, and ii) expression in individuals with and without CVD. It is unclear from the text what the indicated P-values are referring to without the associated figure.

Reply: Thank you for the comment, we completely agree. Both figure and corresponding text are now updated. We believe text now clearly reflects information shown in the figure.

5) How did the authors identify the single genes highlighted as key components of the individual communities? Are they ranked based on fold change in expression? (doesn't appear to be the case for COL1A2).

Reply: Thank you for your comment. The individual genes mentioned in the manuscript (COL1, MYORG and p21) were not selected algorithmically, but are hand-picked as representative of their respective communities (extracellular matrix, mitochondria and p53 signaling, respectively), considering these genes are established components of their respective biological function. Although all the main results of the present study are presented mainly at community-level,

we believe that mentioning few individual genes in the text gives the reader a certain concreteness and facilitates the interpretation of the results. For the interested reader, the data on differential expression, Cox regression and a number of other measures and methods for all individual genes are presented in the Additional file 2 (former Additional file 3).

-In addition, the expression p-values in paragraph line 258-270 don't appear to match those provided in Additional file 3. For example, CDKN1A does not pass the FDR (0.07) and survival analysis p-value is indicated as 0.02 rather than <0.001.

Reply: Thank you for your comment. We apologies for the confusion between the information presented in the manuscript and provided supplement. We chose to report nominal p-value as a metric of the significance in the manuscript but provided only FDR in the supplement for corresponding gene. Supplement is now updated to also include nominal p-values from differential expression analysis.

- Illustrating the magnitude of these gene changes in relation to the rest of the dataset, e.g. highlighting these 3 genes on a volcano plot, would better highlight their importance as differential gene signatures in HF vs. control.

Reply: Thank you for your comment. We agree with the reviewer that using volcano plot to emphasize the extent of change between conditions is important and integral part of differential expression analysis. However, from the point of network-centric analysis that leverages co-expression and network structure we believe this to be redundant.

- Specific data (correlation strength, p-values) for CDKN1A correlation to NT-proBNP and LVEF should also be included in the manuscript if discussed in the results (line 266-268).

Reply: Thank you for your comment. This is now included in the manuscript.

6) The authors use very creative visual representations of the bioinformatics data in the manuscript figures, however these figures (e.g. Fig 4, 5) aren't immediately intuitive to the reader. More details could be provided in the figure legends on the colour/symbol schemes and analyses shown in these graphics. An additional table providing the individual correlation data for each community and clinical variable would also benefit.

Reply: Thank you for your comment. Figures and figure legends are now updated. Additional table with correlations between modules and traits of interest is available within Additional file 2.

Minor comments:

1) Line 189 should indicate that the control group had a higher proportion of female subjects, not male.

Reply: Thank you for your comment. This is not corrected.

2) In figure 3, the authors use a heart schematic as the central illustration. Given that this study is an analysis of skeletal muscle, a different illustration might be more appropriate to convey the study question.

Reply: Thank you for your comment. Figure is now updated.

3) Figure 3, the units for “DEG regulation, patients vs. controls” scale bar should be included, e.g. logFC.

Reply: Thank you for your comment. We believe reviewer was referring to Fig. 4 legend. Figure and figure legend are now updated to correctly reflect the underlying data.

4) Figure 6: panels A and B should be called out separately in the results.

5) Throughout the manuscript, when comparing the disease (HF) cohort to controls, it may be more accurate to describe any changes as they relate to the disease group, rather than the control group. E.g. Line 233 “ECM remodeling community was higher in GTEx controls” should be changed to “was lower in HF patients”.

Reply: Thank you for your comment. This is now revised.

6) Paragraph, Line 236-244: the 4 clinically defined variables used in these comparisons are listed as VO2 peak, daily physical activity, NT-proBNP, and LVEF. From these variables, only 1 network community (p53) has significant correlations with more than 1 clinical variable, not 3 (+ ECM and FA oxidation), as written in the manuscript. When including death/prognosis as a 5th factor, then all 3 communities have 2 or more associations. How this is written, whether “death” is included or distinct from this list of clinical variables, should be revised.

Reply: Thank you for your comment. This is now corrected in the revised manuscript to reflect filtering steps.

7) Line 246: “respectively” should be added to this sentence as low physical activity was only true for ECM, and NT-proBNP for FA oxidation.

8) Line 248: p53 was not associated with daily physical activity in Fig 5, this sentence should be revised.

Reply: Thank you for your comment. We agree with the reviewer that sentences in question are a bit misleading even though we do emphasize overall trends in a data not their significance. The magnitude and corresponding p-values are clearly shown in the corresponding figure. Regardless, this is now corrected in the revised manuscript.

9) Line 296: sentence relates to data in Fig. 4 rather than Fig 3.

10) Line 342: This statement should be supported with references to the appropriate primary research articles.

Reply: Thank you for your comment. This is now revised.

Reviewer #4 (Remarks to the Author):

Rullman et al. present RNA-seq data from human skeletal muscle tissue in patients with HF vs. a variety of control populations. They perform a comprehensive network analysis and identify several gene networks that are dysregulated in association with HF. The study is novel in its use of RNA-seq in skeletal muscle in this population, an organ commonly compromised but little studied. However, the study is limited as it is essentially a 1-experiment analysis without follow-up confirmatory testing or experimental modeling of any of the identified networks. Overall, while interesting and rather novel, the manuscript suffers from poor organization and flow, and challenges associated with the depiction of the data.

Major Comments:

1) Figures and tables are hard to follow as they extend from p28-34 without any listing on those pages for what figure/table is what. Can a label be added to the page (I'm sure this could be deleted in the proofs) facilitate easier review? Similarly, the Additional files are not labeled as such in the files available in the reviewer folder. If these files are improperly titled, perhaps change the tab names for each file to ensure the reviewer knows which additional file is which. Also, the sample sizes for the groups referred to in the text for Additional files 1 and 2 is not stated in the text; this would have also made it easier to identify which file was which. I eventually figured it out but spent more time than was necessary doing so.

Reply: Thank you for the comment and we apologize for the inconvenience.

We understand the reviewer's frustration as we faced the same hurdle on numerous occasions however, final pdf document is not in our control, rather it is automatically generated during submission process based on the submitted data.

Based on the reviewer's suggestion, we have created a PDF document that includes clickable cross-references to all tables and figures within the manuscript. Cross-references to the Additional File 1 and 2 directs to externally deposited materials on figshare and therefore will not be visible within PDF document. We believe this will facilitate easier navigation and revision process.

2) Fig 3 – the images in the circles need to be enlarged; they are rather hard to discern. Also, why are some black and some white? Is this representative of some other kind of difference? If just for contrast, please make them uniform as doing it this way makes it seem like there is a difference (which isn't defined in the legend). More importantly, are these icons validated or the author's self-characterization of the roles of these pathways (e.g., did patients actually meet the clinical definition of cachexia)? What are the checks and x's for? This is not well-described.

Reply: Thank you for your comment. Icons indicating bed rest, exercise, cachexia and senescence are not validated/established in the literature, rather they reflect our own characterization of the related terms. We have now enlarged and updated the icons based on the reviewer's suggestion. We are confident that updated icons together with corresponding explanation within figure legend are self-explanatory and not misleading. We have also updated figure legend to include explanation for the.

In terms of analysis, symbols (checks and x's) reflect results of over-representation analysis between gene communities and genes deemed significantly differentially expressed in bed rest, exercise, cachexia and senescence datasets.

3) I'm sorry, but I simply do not understand what I'm looking at in Figure 5. There appear to be 3 sets of colors but only 2 are defined in the legend...2 of the 3 color schemes are relatively similar; are these supposed to be the same? If so, please make them uniform. Further, while the graphs used are clever and visually interesting, the data in them is too small to read, particularly the statistical comparisons. These need to be enlarged.

Reply: Thank you for your comment. This has now been modified in the revised version. Different shading of the same colors (darker vs lighter) in the Fig. 5 corresponds to the significant vs insignificant segment of the individual figure. We agree with the reviewer this was not clearly explained in the figure legend, and it is now corrected. All text sizes now surpass minimum requirement and correspond to journal guidelines.

We have also updated legend for the Fig.4.

4) Line 238-9 indicates 9 gene communities were significantly correlated with 1 of the 4 clinical variables they studied. As best I can tell, they only show data on the 3 that were correlated with more than 1. These other data should also be presented, even if just in the data supplement.

Reply: Thank you for your comment. We agree with the reviewer that provided information in the manuscripts regarding number of variables included in correlation analysis was misleading. This is now corrected to reflect results presented in the Fig. 4.

Correlation analysis was conducted on the community-level, including clinical variables of interest: NT-proBNP, LVEF, VO_{2peak}, daily physical activity, and prognosis.

In addition to correlation analysis in Fig. 4, we also included information of community-level expression differences between heart failure patients and their respective controls. In Fig. 5 we choose to visually present more in-depth analysis only for the communities with ≥ 2 significant associations. This is to emphasize network communities relevant for the manuscript while results for remaining communities are made available in Additional file 2 (former Additional file 3), under community analysis sheet.

5) First paragraph of discussion, they indicate their data as “describing the mechanisms” but this study is associative not causal (i.e., not definitively showing a mechanism, but rather shows an association between skeletal muscle gene networks and heart failure). They should rephrase this characterization.

Reply: Thank you for your comment. We agree that the observational design of the study wasn't always reflected in the text in the manuscript. This has now been revised.

6) When discussing beta-oxidation, they state, “in a manner consistent with our analysis of the transcriptional signature of long-term bed rest” Is there a study of the transcriptional signature in patients on long-term bed rest? If so, this needs to be cited/referenced. This is partly related to my question about the icons (see #2 above). I'm also confused by the frequent references to bed rest...I'm a HF physician, other than my sickest patients who are hospitalized, HF patients aren't on strict bed rest.

Thank you for the comment. The sentence relates to observation following long-term bedrest in healthy individuals, which serves as an experimental model to study the effects of prolonged physical inactivity. This data, in combination with the analysis of daily physical activity (from accelerometers worn for 7 days) in the heart-failure patients (and controls) serves to test the influence of physical inactivity on gene-expression. To avoid confusion the sentence is now modified to “in a manner consistent with our analysis of the transcriptional signature of long-term bed rest in healthy individuals”

7) The flow of the discussion is a bit abnormal; they spend considerable time on the other pathways in the first half (more than half) of the discussion, only coming to the strongest signal and discussion point (p53 signaling in skeletal muscle) last. I would recommend a significant revision of the flow of this section.

Reply: Thank you for this comment. The idea was to be transparent of the flow chart of the analysis and to integrate the results into a biological context. Nonetheless in accordance with the reviewer comments, the first part of the discussion is now revised and hopefully better balanced.

8) Lastly, there is absolutely no discussion of study limitations, of which there are many.

Reply: Thank you, a study limitation is now included in the revised version.

Minor Comments:

- 1) There are errors of formatting in the references (e.g., ref 4, "Working Group 'Exercise 400 Physiology SC and CR of the IS of C"; ref 10, "Haehling S von" etc...) that should be cleaned up.
- 2) Line 47 "HRrEF"
- 3) There are scattered grammatical and typographical errors.

Reply: Thank you for your comment, this has now been revised.

Reviewers' comments:

Reviewer #1 (Remarks to the Author):

The authors have dismissed my comments and provided a rationale for dismissal that is founded on one citation for a study showing that patients respond to training. The citation and the rationale do not address the issue of a blunted response, which is implied that activity resistance like anabolic resistance. The discussion is centered on the notion of muscles having an inactivity genetic footprint that the authors conclude results from inactivity. However, there are no reliable (quantitative) measurements of daily physical activity in the study, which would be required to support their stance.

The study we referred to is relevant in that it is one of many studies that have shown that, at the level of skeletal muscle, patients with heart failure have a normal response to endurance-type exercise training. However, at the level of central haemodynamics, the impact of such training interventions is very small or even non-existent in some studies which means that the overall improvement in performance and peakVO₂ is also very small. This was actually suggested as early as the 1980s by Franciosa et al. [1] who noted that while cardiac function was the key factor in explaining why heart failure patients had lower aerobic capacity than healthy individuals, it had negligible value in explaining differences within a group of patients—they suggested peripheral factors, i.e. skeletal muscle, as the key to understanding this difference. Subsequently, a number of studies, such as the one by Bengt Saltin et al. we referred to in our last rebuttal, showed that at the level of skeletal muscle, the response to training was retained in heart failure. In fact, this was also reported in the HF-ACTION trial, where patients who responded to exercise with a slight improvement in VO₂ had a substantially better prognosis—this has also been attributed to peripheral adaptations. To our knowledge, the concept of an ‘aerobic resistance’ at the level of skeletal muscle has not been demonstrated in heart failure.

We further do not agree that we conclude the overall difference between patients and controls in our study is explained by physical activity—in fact, our main message is that gene expression not associated with physical activity had the by far strongest association with mortality (and could be an underlying mechanism of anabolic resistance). As for the mitochondrial and ECM-related genes we report: 1. Mitochondrial and ECM-related genes are down-regulated in patients compared with controls. 2. These genes are highly regulated in exercise-training and bed-rest datasets and could therefore differ between the groups due to differences in physical activity. 3. We confirm, through our own accelerometer data, that patients had a substantially lower degree of daily physical activity than control subjects. 4. We further analyzed whether daily physical activity (as measured by accelerometer) was correlated with gene expression and can show that there was a significant positive correlation between daily physical activity and gene expression of mitochondrial and ECM-related genes.

To corroborate this further we have attached two plots showing 1. The difference in PA between patients and controls in our study. 2. The correlation

between PA and ECM-gene expression across both patients and control subjects.

We would argue that if there was a significant difference in the skeletal muscle response to PA patients would display a substantially lower gene-expression at the same level of daily PA but in fact the correlation is very similar and linearly correlated in both patients and controls. This is not in conflict with the concept of anabolic resistance, but it indicates that these transcriptional communities are associated with variations in daily activity to a similar extent in both patients and age-matched control subjects.

Reviewer #2 (Remarks to the Author):

That authors have adequately addressed my previous concerns.

Remaining are a few minor points for revision:

1) It is difficult to correctly associate the various tabs provided in the additional data files to the appropriate section of the text. It would be helpful if the tabs could be numbered or labeled in a way that they could each be called out at the appropriate place in the manuscript text.

Thank you for your comment. We understand the hurdle of navigating between multiple sheets of data however, we were unable to find adequate way to address this issue and confront with journal's template.

2) Figure 4 – The results of enrichment analysis of the gene communities to other clinical conditions (e.g. bed rest, exercise, cachexia, senescence) is provided in Figure 3 and does not necessarily need to be repeated in Figure 4.

Thank you for your comment. We agree with the reviewer that we are in fact presenting results of the enrichment analysis across both figure panels but due to the layered complexity of the analysis we wanted to facilitated interpretation of the result section by emphasizing relevant information.

3) Discussion Line 364: “suggesting that these communities are plausible key processes in the pathophysiology of skeletal muscle in heart failure”. This statement should be revised. That these “may be important signatures of skeletal muscle transcriptional remodeling in HF” would be more appropriate, as there is no data demonstrating an intrinsic functional decline in the skeletal muscle of these patients in the present study that can be correlated with these signatures.

Thank you for your comment. This is now corrected in the revised manuscript to more correctly reflect our findings.

Reviewer #3 (Remarks to the Author):

This is a revised manuscript. The authors have addressed my biggest concern regarding flow and readability, and the manuscript is far more readable now. Of note, they did not (as indicated in the rebuttal letter) bring the p53 story to the forefront of the discussion. However, the extensive changes make the flow substantially better, obviating the need for that revision. I have the following comments that should be addressed, but represent rather minor changes that can be addressed quickly without further experimental testing/analysis.

Major Comments:

1) I have 2 comments regarding the new limitations section. (1) in the first paragraph, they indicate that their design makes the validity “likely high” – this is subjective and biased wording and isn’t definitively proven by the data in the manuscript. I’d like to see this sentence tempered to something more like, “We used ...internal/external...to improve/strengthen the validity of our observations.” (2) One last but important limitation is that the other cohorts used for comparison would have had sequencing by different methodologies/different facilities. This makes them suboptimal controls – I totally understand why they were used (this is very hard tissue to obtain), but this needs to be acknowledged as a source of potential/probable bias.

Thank you for your comments. (1) This is now corrected in the revised manuscript. (2) Combining samples from different studies can introduce biases (sequencing technologies, sequencing depth, lab site and lab conditions, lab technician/s, etc.) which if not addressed properly can have major impact on the downstream analysis, leading to false discoveries and misleading conclusions. As stated in the method section, we used ComBat-seq in combination with PCA analysis to address this and to evaluate the impact of the batch correction on the sample grouping. Despite employing one of the leading algorithms in the field, we acknowledge that in current setting where true biological signal is confounded with the technical bias this remains a challenge and should be addressed. This is now added in the revised manuscript.

Minor comments:

1. IQR is not defined at first use. I presume it means interquartile range (and that use is appropriate in the first section of the results. However, the first use is in the methods pertaining to the STRRIDE genes and I do not understand what they mean by identifying genes with $IQR > 2$. Is this a typo in this spot?

Thank you for your comment. We apologies for the confusion, this is now revised.

Gene filtering based on IQR cutoff is a common preprocessing step in microarray analysis. This step facilitates the removal of uninformative genes, thus reducing signal-to-noise ratio and increasing the power to detect truly differentially expressed genes. Furthermore, exclusion of low-variance genes is important for network construction, as it help minimize spurious correlations between genes[2].

2. For the new Additional Table 1, I suggest updating the “Heart Failure (N=58)” column header to something like “Study Patients (N=58)” as this would be clearer.

Thank you for your comment. Additional Table 1 and 2 (now Supplementary Tables) are now updated based on the suggestion.

1. Franciosa, J.A., M. Park, and T.B. Levine, *Lack of correlation between exercise capacity and indexes of resting left ventricular performance in heart failure*. *Am J Cardiol*, 1981. **47**(1): p. 33-9.
2. Chockalingam, S., M. Aluru, and S. Aluru, *Microarray Data Processing Techniques for Genome-Scale Network Inference from Large Public Repositories*. *Microarrays* (Basel), 2016. **5**(3).